**Aerosol immission maps and trends over Germany with hourly data at four rural**
**background stations from 2009 to 2018**
Jost Heintzenberg[1], Wolfram Birmili[2], Bryan Hellack[2], Gerald Spindler[1], Thomas Tuch[1], and
Alfred Wiedensohler[1]
1: Leibniz Institute for Tropospheric Research (TROPOS), Permoserstr. 15, 04318 Leipzig,
Germany
2: German Environment Agency, Wörlitzer Platz 1, 06844 Dessau-Roßlau, Germany
Abstract
Ten years of hourly aerosol and gas data at four rural German stations have been combined
with hourly back trajectories to the stations and inventories of the European EDGAR emission
database yielding immission maps over Germany of $PM_{10}$, particle number concentrations, and
equivalent black carbon (eBC).  The maps reflect aerosol emissions modified with atmospheric
processes during transport between sources and receptor sites.  Compared to emission maps
strong Western European emission centers do not dominate the downwind concentrations
because their emissions are reduced by atmospheric processes on the way to the receptor area.
$PM_{10}$, eBC, and to some extent also particle number concentrations are rather controlled by
emissions from Southeastern Europe from which pollution transport often occurs under dryer
conditions.  Newly formed particles are found in air masses from a broad sector reaching from
Southern Germany to Western Europe which we explain with gaseous particle precursors
coming with little wet scavenging from this region.

24        Annual emissions for 2009 of $PM_{10}$, BC, $SO_2$, and $NO_x$ were accumulated along each

trajectory and compared with the corresponding measured time series.  The agreement of each
pair of time series was optimized by varying monthly factors and annual factors on the 2009
emissions.  This approach yielded broader summer emission minima than published values that
were partly displaced from the midsummer positions.  The validity of connecting immission
and emission of particulate pollution was tested by calculating temporal changes of eBC for
subsets of back trajectories passing over two separate prominent emission regions, region A to
the Northwest and B to the Southeast of the measuring stations.  Consistent with reported
emission data the calculated immission decreases over region A are significantly stronger than
over region B.

## 1 Introduction

The atmospheric aerosol is known to influence the Earth's radiation budget because it directly scatters and absorbs solar radiation (Schwartz, 1996; Bond et al., 2013), and acts as cloud condensation nuclei, thus modulating the optical properties and lifetimes of clouds (Twomey, 1974; Penner et al., 2004). In many regions of the globe that had undergone industrialization early on, anthropogenic aerosol concentrations are currently in decline (Leibensperger et al., 2012; Zanatta et al., 2016). With respect to declining concentrations and emissions, Samset al. (2018) suggest that removing present-day anthropogenic aerosol emissions – assuming constant greenhouse gas emissions, could lead to a global mean surface heating as high as 0.5–1.1°C.

Besides climate, the atmospheric aerosol has been acknowledged to influence human health through respiratory and cardiovascular health endpoints (Anderson et al., 2012). Lelieveld et al., (2015) quantified the world-wide burden of disease (premature mortality) due to outdoor pollution, large part of which was attributed to airborne particulate matter. It is apparent that the distribution of adverse health effects is very uneven among the world-wide population, depending on the local level of outdoor pollution.

In view of the described man-driven effects it seems imperative to develop instruments to reliably monitor changes in anthropogenic aerosol concentrations as well as an understanding of the balance between aerosol sources and measured concentrations. Researchers have strived to obtain a spatial picture of the distribution of pollutants, and to achieve a connection between the sources of pollution and concentrations downwind. A widely used method has been the extrapolation of concentrations measured in one or several locations into two-dimensional space through the use of meteorological dispersion approaches: The first maps of particulate

air pollutants over Europe were constructed in the 1970s with the help of coarse emission data
and simple trajectory models (Eliassen, 1978). Statistical methods were developed to connect
pollution sources and ensuing aerosol concentrations at receptor sites (Miller et al., 1972;
Friedlander, 1973; Cass and McRae, 1983). By combining statistics with back trajectory data
sectorial information about sources controlling the composition of the aerosol over Southern
Sweden was derived by Swietlicki et al., (1988). Later the approach of using back trajectories
to map aerosol sources was refined by Stohl (1996) and tested with one-year sulfate data from
the co-operative program for monitoring and evaluation of the long-range transmission of air
pollutants in Europe (EMEP, www.emep.int). In a similar approach with five years of aerosol
data from a single Siberian receptor site Heintzenberg et al. (2013) identified potential source
regions over Eurasia and with aerosol data from four Swedish icebreaker expeditions over the
Central Arctic (Heintzenberg et al., 2015). Charron et al. (2008) constructed concentration field
maps to identify the source regions of specific types of aerosol particle size distributions
arriving in England. All these works share the approach that time-dependent information on
concentrations measured at receptor site(s) are transformed into space, thus allowing
conclusions on the potential source regions of gaseous and/or particulate emissions.

With more comprehensive air quality models concentrations of specific aerosol were
mapped over Europe together with short temporal developments (e.g., Schell et al., 2001). For
specific episodes high spatial resolution aerosol concentration maps in urban and non-urban
European areas have been generated with sophisticated chemistry transport models (e.g.,
Beekmann et al., 2015; Riemer et al., 2004; Wolke et al., 2004). For the years 2002 and 2003
Marmer and Langman (2007) analyzed the spatial and temporal variability of the aerosol
distribution over Europe with a regional atmosphere-chemistry model. They found that
meteorological conditions play a major role in spatial and temporal variability in the European
aerosol burden distribution. Regionally, year to year variability of modeled monthly mean
aerosol burden reached up to 100% because of different weather conditions.

In the present study ten years of hourly aerosol data at four German stations were available
for the identification of potential source regions. As it appears unrealistic to analyze such a
large database with advanced chemical transport models we resorted to the well proven
approach of utilizing back trajectories cited above and connected the results to emission fields.
We define the resulting concentration maps of particulate and gas parameters as immission
maps because they represent long-term average emissions of air pollutants modified by the
controlling atmospheric processes along the pathways to the receptor sites. In Charron et al.
(2008) this approach is termed "concentration field map method". With a much larger data set
spanning a much tighter network of 1500 stations Rohde and Muller (2015) used the Kriging
interpolation approach (Krige, 1951) to construct air pollution maps over China. Another
approach to construct pollution maps over the province Henan, China was used by Liu et al.,
(2018). They combined an emission inventory with chemical modeling and back trajectories
to derive high resolution maps of particulate and gaseous pollution components and find that
emissions from neighboring provinces are important contributors to local air pollution levels.

Recent political, economic and technological developments in Europe have caused
substantial changes in the emission of air pollutants. Lavanchy et al. (1999) deduced a trend in
atmospheric black carbon from preindustrial times to 1975. Strong downward trends in major
aerosol components before and after the German reunification (1983-1998) over rural East
Germany were reported by Spindler et al., (1999). For the years 2003 – 2009 Kuenen et al.,
(2014) published trends in the development of aerosol emissions as elaborated from reported
emissions. The German Environmental Agency (GEA) publishes trends in air pollution as
measured at a number of ca. 380 federal and state air quality stations (Minkos, 2019).
According to these records, $PM_{10}$ mass concentrations declined by approximately 25 % over
the period 2000-2019

Combining long-term aerosol and gas data at the four stations of the present study provide
an excellent data base for identifying both the most important source regions and possible
temporal changes. During the ten recent years covered by our data we expected noticeable
systematic changes in our time series that can be interpreted in terms of emissions. As a side
result in the process of deriving long-term emission trends of major air pollutants over Germany
information of the monthly disaggregation of annual aerosol emissions can be derived.


**2  Aerosol and trace gas data**

The core data of the present study have been measured at the stations Melpitz (ME),
Neuglobsow (NG), and Waldhof (WA) of the German Ultrafine Aerosol Network GUAN
network (Birmili et al., 2016) and at station Collmberg (CO) operated by the Saxonian
Environment Agency. These four rural background stations lie in the northeastern lowlands of
Germany at distances between 30 and 205 km from each other. Ten-year-average particle mass
concentrations under 10 μm particle diameter ($PM_{10}$) and their standard deviations at the four
stations are rather similar: 15±13, 22±12, 14±10, and 15±11 $\mu gm^{-3}$ at CO, ME, NG, and WA,
respectively. The corresponding long-term average particle number concentrations between 10
and 800 nm particle diameter ($N_{10-800}$) and their standard deviations at the three GUAN-stations
are 5400±4100, 3600±2300, and 4300±2800 $cm^{-3}$, respectively. Basic statistics on particle
number and eBC mass concentrations of the three GUAN-stations were presented in Sun et al.
(2019) whereas details about instrumentation and their maintenance can be found in Birmili et
al., (2016). The ensemble of hourly data at the four stations is the base of the pollution maps
derived in this work.

TROPOS-type mobility particle size spectrometers (MPSS, Wiedensohler et al., 2012) were
used to record particle number size distributions across the particle size range 10-800 nm.
Quality assurance of the long-term measurements followed the recommendations of
Wiedensohler et al. (2018) including weekly inspections as well as monthly and annual
maintenance intervals.  Once a year the MPSS were intercompared against a reference MPSS
of the WCCAP (World Calibration Center for Aerosol Physics) either on-site and/or at the
calibration facility.  The lower detection limit of the MPSS is around 30 cm$^{-3}$ for a time
resolution of 30 minutes. Equivalent Black Carbon (eBC) was determined by multi-angle
absorption photometers (MAAP) using a mass absorption cross section of 6.6 m² g$^{-1}$ (Petzold
et al., 2013; Nordmann et al., 2013; Birmili et al., 2016).  An intercomparison of multiple
MAAP instruments resulted in an inter-device variability of less than 5% (Müller et al., 2011).
While the MAAP deployed at the TROPOS station Melpitz was biannually compared to the
reference absorption photometer at the WCCAP in Leipzig, the instruments at the UBA stations
Waldhof and Neuglobsow were serviced by the manufacturer.  For hourly measurements of
PM$_{10}$ continuous oscillating microbalances (Thermo Scientific TEOM 1400) were utilized at
stations CO, NG, and WA.  At station ME PM$_{10}$ was determined in daily filter samples (0:00
to 24:00 CET), Spindler et al. (2013).  The TEOM1400-instrument and gravimetric filter
sampling are different methods for particle mass concentrations.  The TEOM collects
particulate mass on a vibrating substrate (tapered element) and registers the change of the
oscillation frequency that is decreasing with mass loading (Patashnick and Rupprecht, 1991).
The TEOM operates at a constant temperature setting above ambient (typically 30– 50°C) to
prevent contraction and expansion of the tapered element and reduce interferences from water
vapor condensation.  However, heating the ambient air enhances volatilization of particle-
bound semivolatile compounds (e.g., ammonium nitrate and some organic species) resulting in
an underestimation of PM when semivolatile material dominates the particulate phase during
cold seasons. The condensation and evaporation of ammonium nitrate and organic species can
also influence the filter sampling under ambient conditions. Here the effect can be balanced
partly by the temperature variation during the daily filter sampling. However, the results of both
methods mostly are in good agreement (e.g., Zhu et al., 2007).

Hourly aerosol data from the three GUAN-stations during 2009 – 2015 (NG ≥2011) have
been utilized in a previous study (Heintzenberg et al., 2018) to understand aerosol processes
during air mass transport between the stations. In the present study the data set was enlarged
to include the additional station Collmberg and data at all stations from the year 2016 through
2018. The integral aerosol parameters particle number concentration ($N_{10-800}$, $cm^{-3}$), light
absorption-equivalent mass concentration of Black Carbon (eBC, $\mu gm^{-3}$), and particle mass
concentrations under 10 µm particle diameter ($PM_{10}$, $\mu gm^{-3}$) were utilized. $N_{10-800}$ is based on
the integral over measured particle size distributions from 10 to 800 nm.

$NO_x$ and $SO_2$ emitted by anthropogenic combustion processes are transformed in the
atmosphere and add to the anthropogenic aerosol. At the three GUAN stations both are
measured with the same temporal resolutions as the aerosol data. Additionally, at Collmberg
$NO_x$-data could be utilized in the interpretation of the aerosol data. The trace gas analyzers for
$NO_x$ and $SO_2$ were calibrated with test gases for NO (NO in $N_2$) and $SO_2$ ($SO_2$ in $N_2$, both Air
Liquide, Germany). $NO_2$ was produced in a gas phase titration device (GPT APMC370,
Horiba, Germany) by quantitative oxidation of NO test gas (Rehme, 1976). The trace gas
analyzers were used in an optimal range and all registered values (also below the detection
limit) were used for this long-term study. As most particle numbers in polluted continental
environments tropospheric ozone is a secondary atmospheric pollutant. We utilized hourly
ozone data taken at all four stations throughout the studied time period as ancillary information
in the discussion of particle-number related results. For the ozone measurements a common
trace gas ozone monitor was used (Horiba APOA-350). This device quantifies tropospheric
ozone by UV Absorption and use the cross-flow modulation principle. Ambient air with and
without ozone (elimination by a selective scrubber) was used alternatively in the measuring
cuvette yielding a very stable ozone signal. The devices were calibrated using an ozone-
standard (Ozon-Calibrator, Thermo Environmental Instruments 49PS).

Table 1 gives an overview over the instrumental characteristics of all stations and the total
number of validated data hours for each utilized component. The minimum is 57962 hours for
validated MPSS-data at the three GUAN-stations and the maximum with 88838 validated data
hours for $NO_x$ at all four stations. Strictly concurrent (by the hour) are less validated data hours.
For MPSS, eBC, and $SO_2$-data at the GUAN-stations this numbers is 48533 hours, and 48114
and 47729 hours for $PM_{10}$ and $NO_x$-data, respectively, at all four stations. However, these
reduced strictly concurrent numbers do not substantially affect the 10-year-average maps
discussed below.


**3 Back trajectories**

With the HYSPLIT4 model (Stein et al., 2015) and based on the meteorological fields from the
Global Data Assimilation System with one-degree resolution (GDAS1,
https://www.emc.ncep.noaa.gov/gmb/gdas/) three-dimensional trajectories were calculated
arriving every hour at a height of 500m above ground level at the four stations. The trajectories
were calculated backward for up to five days using the meteorological fields from the server at
Air Resources Laboratory (ARL), NOAA (http://ready.arl.noaa.gov), where more information
about the GDAS dataset can be found.  In the immission maps constructed with extrapolated
measurements at the stations and in any comparisons with emissions along the back trajectories
only trajectory points under 1000 m altitude above ground were utilized.  Turbulent
atmospheric mixing is included in parameterized form in HYSPLIT4.  The present study
utilizes the default version of this parameterization according to Draxler and Hess (1998).  The
back trajectories are calculated with the base version of HYSPLIT4 that does not include any
specific dispersion and scavenging of atmospheric trace substances.  Precipitation along the
trajectories was used in the interpretation of the immission maps.  The precipitation values
mapped in the present study and the temperature values used in the trend discussion of $N_{10\text{-}800}$
are those listed by HYSPLIT4 at each point of a trajectory.  They are meteorological parameters
at the nearest grid cell of the assimilated global meteorological fields provided by the U.S.
National Weather Service's National Centers for Environmental Prediction (NCEP)
(Kanamitsu, 1989).  Average horizontal wind speeds in between two one-hour trajectory steps
were calculated from the distance covered by a trajectory between two successive steps.  With
the 350593 hourly back trajectories from the four stations the time series of $N_{10\text{-}800}$, $PM_{10}$, and
eBC were extrapolated over Germany and part of the neighbor countries.  At Melpitz $PM_{10}$-
data were only available as daily averages.  Thus, the daily average concentrations were
extrapolated along each hourly trajectory of the respective day.


**4  Emission data**

For the interpretation of the immission maps we used the emission data set version 4.3.2 for
2009 of the components particle mass concentrations below 10 μm ($PM_{10}$), BC, $NO_x$ and $SO_2$
as compiled in the Emissions Data Base for Global Atmospheric Research (EDGAR,
https://edgar.jrc.ec.europa.eu/overview.php?v=432_AP, DOI (https://data.europa.eu/doi/10.29
04/JRC_DATASET_EDGAR). This data set concerns primary emissions only and has been
introduced by Crippa et al., (2018). All human activities, except large scale biomass burning
and land use, land-use change, and forestry are included in the data base. Emissions of coarse
particles from agricultural surfaces are not included. They are, in fact, very sensitive to soil
and weather conditions, and thus not trivial to quantify. Primary aerosol emission data are
generally characterized by rather high uncertainties. For the EDGAR data base Crippa et al.
(2018) report a range of variation in 2012 between 57.4% and 109.1% for $PM_{10}$, and between
46.8% and 92% for BC. Even higher uncertainties in PM emissions might come from super-
emitting vehicles that are not considered in this data base (Klimont et al., 2017). In our maps
and trend calculations we applied the grid values of emission data that were listed in the
EDGAR inventories no more than 30 km away from any trajectory time step.


**5  Results and discussion**
**5.1    Aerosol concentration maps (immission maps**
The trajectory-extrapolated $N_{10-800}$, $PM_{10}$, and eBC from the four stations yielded immission
maps averaged over the period 2009 – 2018, that are collected in Figs. 1-2. Both, the particle-
number related $N_{10-800}$ and the particle-mass related $PM_{10}$, and eBC exhibit systematic seasonal
variations. Most events of new particle formation (NPF) over the continents occur during the
photochemically active summer months (Kulmala et al., 2004) whereas the particle-mass
related aerosol parameters due to combustion processes exhibit highest concentrations during
the winter months (Matthias et al., 2018). Consequently, we constructed two maps for each
discussed component: One of averages over the months April through October and one of
averages over the months November through March. Only map cells with at least 300 trajectory
hits are discussed. Interpreting these hits in terms of Poisson-statistics would then yield a
maximum uncertainty of 5.8% per cell. In terms of a Gaussian statistic the arithmetic cell-
averages displayed in the maps exhibit standard deviations of cell averages that are less than
six percent.

The maps of $N_{10-800}$ in Fig. 1 show distributions of air masses over Germany and adjacent
countries related to particle numbers instead of particulate mass. There are two arguments for
showing maps of number related results. First, particle number concentrations are connected
with cloud processes, their formation (Pruppacher and Klett, 1978), radiative effects, e.g.,
albedo (Twomey, 1974), and precipitation (Li et al., 2011). Second, in the area of aerosol-
health issues ultrafine particles (< 100 nm diameter) have been gaining attention in recent years
(Wichmann and Peters, 2000), i.e. an increasing number of health effects is attributed rather to
particle number than to particle mass. The fact that NPF-events occur concurrently in or near
the top of the continental planetary boundary layer over wide geographical regions (e.g.,
Wehner et al., 2007) is partly due to concurrent advantageous photochemical conditions
allowing for the formation of condensable vapors, in particular global radiation (Birmili et al.,
2001). Two other factors constraining NPF are the availability of gaseous particle-precursors
and the concurrent pre-existing aerosol.

The summer map (4-10) of $N_{10-800}$ exhibits the high values in the Southwest-to-Northeast-
sector of the map. Highest values are concentrated in a belt reaching from Burgundy through
Switzerland, Southern Germany, Czech Republic to Southwestern Poland. Interestingly, this
belt of high $N_{10-800}$ is collocated to large extent with a belt of high summer ozone concentrations
(cf. Fig. S1). This photochemically controlled pollutant (Monks et al., 2015) exhibits highest
summer concentrations in air masses from Southwestern Poland and Northern Czech Republic,
a region from which high ozone values are reported (Struzewska and Jefimow, 2013; Hůnová,
2003; Hůnová and Bäumelt, 2018).  However, the summer map of $N_{10-800}$ does not show the
highest values in air masses from the region with highest ozone pollution.  High particle
numbers in air masses coming over the Alps from Northern Italy may be related to the high
emissions of air pollutants in the Po Valley that are known to reach frequently through so called
alpine pumping (Winkler et al., 2006; Lugauer and Winkler, 2005; Reitebuch et al., 2003) over
the mountains.  The high $NO_x$-concentrations in air masses from Northern Italy in both summer
and winter maps (see Fig. S2) indicate that pollution from south of the Alps can even reach
Northeastern Germany.  In the winter map of $N_{10-800}$ (11-3 in Fig. 1) the belt of highest summer
values is apparently complemented by more transalpine pollution transport and by transport
from the Southeast.  The lower photochemical activity in winter is reflected in the lower winter
ozone concentrations in Fig. S1, albeit transalpine pollution transport is still visible in the winter
map of $NO_x$ in Fig. S2.  Northwestern Italy also shows up as an emission hot spot in the maps
of trajectory-summed emissions in Fig. S4.

In both summer and winter the maps of $PM_{10}$, and eBC in Fig. 2 exhibit a clear Northwest-
to-Southeast structure with the cleanest sector being in the Northwest covering the coastal area
of the North Sea, the BENELUX countries Belgium, the Netherlands, and Luxemburg, and
Northwestern Germany.  The strongest contrast between the cleanest Northwesterly and the
most polluted Southeasterly map sectors is seen in the winter map of eBC.  Highest average
concentrations are measured in airmasses from the Southeastern half of the map, most strongly
expressed in $PM_{10}$ and eBC with maxima in a region leading from Southwest Poland through
the Czech Republic, Slovakia, Austria, and former Yugoslavia to Northeastern Italy.  The back
trajectories in the Southeastern sector of the maps for $PM_{10}$ and eBC point towards countries,
in which the emissions of air pollution in the past 20 years developed very differently as
compared to those in Western Europe.  According to the European Environment Agency
(https://www.eea.europa.eu/data-and-maps/dashboards/air-pollutant-emissions-data-viewer-2)
the latter parts of Western Europe experienced a strong and nearly monotonous decrease in
emissions of $PM_{10}$ whereas the emissions in Poland, Czech Republic, Slovakia, Austria, former
Yugoslavia, and Italy stayed nearly constant or even increased in recent years after the dramatic
decreases in the course of the political developments of the 1990ies. The seasonal maps of the
combustion derived $SO_2$ in Fig. S3 look very similar to the those of the particle-mass related
maps of $PM_{10}$ and eBC, again the strongest NW/SE-contrast visible in winter.

5.2. Pollutant emissions and atmospheric processes

In Fig. 3 annual average emissions of $PM_{10}$, BC, $SO_2$, and $NO_x$ are mapped for 2009 according
to the EDGAR emission database. Except for the absolute numbers the maps for $SO_2$, and $NO_x$
look rather similar to those for particulate emissions. They all emphasize highly populated and
industrialized emissions center. Beyond that the $SO_2$-map accentuates individual large
combustion sources such as conventional power plants. Whereas the strong emissions in
Northern Italy are seen in the maps of $PM_{10}$, BC, and $NO_x$ emissions in the countries in the
Southeastern sector of the maps by no means reflect the high concentrations of particulate
components seen in the immission maps of Figs. 1 and 2.

The seeming discrepancy between the immission maps in Figs. 1 and 2 and the emission
maps of Fig. 3 can be resolved. For that purpose, the EDGAR-emissions of $PM_{10}$, BC, $SO_2$,
and $NO_x$ along all 350593 hourly back trajectories to the four stations during the ten studied
years were summed up. Then the sums were extrapolated back along each trajectory. In Fig.
S4 10-year average maps of these extrapolated emission sums are displayed. As in Fig. 3 except
for the absolute numbers there is a strong similarity between the four mapped component sums.
Because of the integral nature of the mapped results one cannot expect the maps in Fig. S4 to
locate correctly specific emission centers. However, they certainly indicate the map sectors
from which the most substantial emissions could have reached the stations. As in Figs. 1 and
2 the Southeastern sectors of the maps of integrated emissions most prominently show up.
Interestingly, the maps in Fig. S4 also indicate the highly polluted region of Northwestern Italy
(Diémoz et al., 2019a; Diémoz et al., 2019b). The emissions from the emission centers in
Northwestern Europe are hardly discernible in Fig. S4. They do show up (most strongly in Fig.
S4c for $SO_2$-emission sums) as apparent emissions over the adjacent North Sea. We interpret
the "misplaced" emissions over the North Sea as air mass transport from the North Sea via the
emission region in the BENELUX countries to the receptor sites that was not compensated by
other low pollution air transport from the North Sea to the stations that had not passed over the
Northwestern European emission centers.

Two major atmospheric processes will reduce the concentrations of emitted or in situ formed
aerosol particles: dilution through mixing with cleaner air masses and wet scavenging through
in-cloud and sub-cloud processes. As a tracer of the first of these two processes Fig. 4a gives
the long-term average geographical distribution of trajectory derived wind speed over the study
area. Highest average wind speeds and ensuing atmospheric mixing is seen over the major
emission centers of Northwestern Germany, the BENELUX countries and adjacent seas
whereas lowest wind speeds are seen over Northern Germany and the Southeastern neighbor
countries. The long-term average geographical distribution of precipitation as taken by
HYSPLIT from the GDAS meteorological fields in Fig. 4b corroborates the results about
atmospheric cleaning processes indicated in Fig. 4a. The small absolute numbers in Fig. 4b are
due to the episodic nature of precipitation: most of the time it does not rain or snow. The blue
crescent reaching from the North Sea through the BENELUX countries, Eastern France,
Switzerland and the alpine region exhibits maximum precipitation values while Southern and
Eastern Germany with the adjoining countries to the East and Southeast show minimum
precipitation values. Thus, in the long term we expect much of the high Western European
emissions to be scavenged to a substantially by wet processes. In addition, air masses arriving
from Western and Northwestern directions at the stations usually cross the Western European
emission centers with much lower pollution burdens than air masses coming from the polluted
countries of Southeastern Europe arriving at the corresponding map borders (cf. Fig. $PM_{10}$ —
36th maximum daily average value in μg m$^{-3}$, 2005 in EEA, 2009).

**5.3. Immission trends for air from specific source regions**

As mentioned in the introduction, the pollutant emissions reported by the European and national
Environment Agencies represent a synthesis of known pollutant sources combined with
assumed emission factors. These emissions are typically used as input for air quality modelling
and subsequent assessment, as well as for trend analyses. However, it remains unclear to what
extent these reported emissions are realistic, and whether their trends represent the trend in true
emissions. Here, we attempt to assess spatially-resolved trends in real particulate emissions by
an analysis of measured concentrations (immissions) in air masses travelling over source-
specific regions.

To test our method, we selected two pronounced source regions in Europe, located within
1000 km distance from our observation sites. These regions were defined by emission hotspot
regions that can be seen in the EDGAR emission maps in Fig. 3a-b and comprise: Region A
(Be-NL-NRW; comprising most of Belgium, southern parts of the Netherlands, and much of
the German state North Rhine-Westphalia) and Region B (CZ-PL-SK; comprising the central
parts of the Czech Republic, southern parts of Poland, and adjacent areas of Slovakia.)
According to the European Environment Agency (EEA) these are regions, where reported
particulate emissions have developed differently during the past 10 years. Our goal is to verify
this through an analysis of real atmospheric observations over this period.
Temporal trends were computed using the customized Sen–Theil trend estimator (Sen, 1968;
Theil, 1992). The Sen–Theil estimator is the median of many slopes calculated in a continuous
or non-continuous time series, with its robustness against outliers being one of its main assets.
For the detailed description of this trend estimator we refer to Sun et al. (2020), Section 2.3.1.
Here we computed the Sen–Theil estimator for hourly observation data at stations ME, NG,
and WA. Subsets of back trajectories were selected that spent at least 1, 3 , 6, or 12 hours over
the source regions A and B. Depending on that criterion, different sub-sets were analyzed. The
difference in median eBC mass concentration between air masses arriving from source region
A and B is obvious, as could already be determined in the corresponding immission maps (Fig.
2c-d). As we learned from Sect. 5.2 these immission maps are strongly influenced by the
different meteorological conditions governing atmospheric dispersion in different wind
direction, so that these values allow no direct conclusion on the strength of emission sources
located upwind.

We analyzed the temporal trends in eBC over the period 2009-2018 for the subsets belonging
to Regions A and B – assuming that these systematic differences in meteorological conditions
should even out over such long observation periods. Table 2 shows that the Sen–Theil slope
estimator for Region A is between -7.6 % and -5.1 % for the three observation sites and the
requirement of a back trajectory to have spent at least 6 hours over Region A. For region B,
the corresponding Sen–Theil slope estimators are between -4.0 % and -2.7 % for the
observation sites. As we can read from these results, the annual decrease in eBC is more
pronounced for air masses that have travelled over Region A.

Between 2009 and 2017 for the EU member states of Belgium, the Netherlands, Germany,
the Czech Republic, Poland, and Slovakia the annual rates of decrease in reported emissions
were between -4.9 and -6.1 % for the first three countries, and between +0.5 and -2.8% for the
latter    three    (https://www.eea.europa.eu/data-and-maps/dashboards/air-pollutant-emissions-
data-viewer-2).  As compiled in Table 2 these reported trends are largely consistent with the
rates of changed derived from our eBC immission trends.  Although we need to keep in mind
that the six nation states only partially contribute to our regions A and B, it seems valid to
conclude that BC emissions in region A indeed decreased more rapidly in the past decade
compared to region B.  Our approach seems able to differentiate between concentrations trends
in air masses that have passes over rather different source regions.  This might represent a step
towards the assessment of changes in real-world emissions allocated in specific source regions
over multi-annual periods.

**5.4. Comparison of immission and emission trends**

Besides the map comparison a second approach was used to connect emission data with the
measured aerosol time series.  Along each of the hourly back trajectories the emissions
according to the EDGAR database were summed up.  Then monthly medians of the emission
sums and the measured parameters were formed.  The EDGAR database reports annual average
emissions.  $PM_{10}$, black carbon and other combustion related air pollutants show substantial
annual variations with high winter and low summer values at non-urban sites (e.g.,
Heintzenberg and Bussemer, 2000).  In emission modeling the temporal variation of annually
reported emissions is considered by disaggregating the annual values with monthly, weekly and
daily factors (Matthias et al., 2018).  For the time-resolved comparison of $PM_{10}$ and BC-
emissions with $PM_{10}$ and eBC-concentrations at the GUAN-sites monthly medians of $PM_{10}$ and
eBC-values at the stations were formed and plotted in Fig. 5.  We expected both, seasonal
variations and a long-term trend in the emissions. For $M$ hours per month of measured
components at the four stations the annual average EDGAR-emissions $E_{PM10}$, $E_{BC}$, $E_{SO2}$, and
$E_{NOx}$ were summed up along the 121 trajectory steps leading to the stations. Then monthly
medians $\tilde{E}_{i=1,4}$ were formed according to Eq. 1 (exemplified for BC). Medians were chosen to
reduce the effect of outliers due to local emission and scavenging events.

$\tilde{E}_{BC} = Median(\sum_{n=1}^{121} E_{BC})_{m=1,M}$                      Eq. 1

The monthly median emission sums $\tilde{E}_{i=1,4}$ were modified with a monthly ($f_m$) and an annual
factor ($g_y$) in order to simulate respective median monthly measured concentrations taken over
all stations. Thus, for each component 12 monthly and 10 annual trend factors determined the
agreement of modified summed emissions and measured concentrations. As objective or utility
function $\chi^2$ the sum of squared deviations between annually and monthly modified emission
sums and monthly median measured concentrations was formed taken over the 120 months of
the present study (exemplified for BC in Eq. 2).

$\chi^2_{BC} = \sum_{j=1}^{120}\left(f_{m=1,12} \cdot g_{y=1,10} \cdot \tilde{E}_{BC} - eBC\right)^2$                      Eq. 2

$\chi^2$ was minimized with a Generalized Reduced Gradient (GRG) solver (Lasdon et al., 1978)
that optimized the12 monthly and 10 annual factors for each of the four measured components.
We used Excel's® implementation of the GRG-solver procedure for the optimization. After
optimizing month and trend factors the average relative deviation between emission-simulated
and measured monthly median curves is 14%, 21%, 25%, and 18% for $PM_{10}$, eBC, $SO_2$, and
$NO_x$, and respectively. The optimized monthly median emission sums for all four parameters
are displayed in Fig. 5 together with the measured monthly median concentrations.

A ten-year trend in emissions of $PM_{10}$, BC, $SO_2$, and $NO_x$, and average monthly factors for
the respective parameters are the two essential results derived from the optimization approach.
The ten-year trends relative to 2009 are collected in Fig. 6. Annual averages of the relative
differences between the monthly median measured parameters and the corresponding emission
derived parameters were formed and applied to the GUAN-trend values displayed in Fig. 6.
The resulting error bars on the trends serve as estimates of the uncertainties of the optimization
approach. The general trend in Fig. 6 is downward to minima between 30 and 70% of the 2009
values in 2016/17 after which all parameters exhibit increases, most strongly $PM_{10}$. $SO_2$ shows
the strongest decrease whereas $PM_{10}$ and $NO_x$-emissions diminished the least. In 2010/2011
the trend curves of $PM_{10}$ and $NO_x$ in Fig. 6 show a slight increase that can be linked to a recovery
of economic activity after the world-wide financial and economic crisis during the period 2007-
2009.    The increase in $PM_{10}$ is also visible in the trend curves relative to
2005 published by the German Environment Agency
(https://www.umweltbundesamt.de/daten/luft/luftschadstoff-emissionen-in-
deutschland/emissionen-prioritaerer-luftschadstoffe).

The results of two comparisons of our trends with data reported by the German and European
Environment Agencies are added to Fig. 6. In general, the trends reported by the German
Environment Agency for all German emissions exhibit weaker reductions than the results of
the present study. Only for $PM_{10}$ in 2011 and 1013 the present study yields higher values than
GEA. We note that primary $PM_{10}$-imissions may have substantial contributions from wind
erosion of agricultural soils (Panagos et al., 2015) that are not incorporated in present
anthropogenic inventories. $SO_2$ exhibits the strongest trend discrepancies with much stronger
reductions in trend of the present study as compared to GEA results. As Germany has been
reducing $SO_2$ emissions systematically since the nineteen eighties one would not expect any
further strong trends during the time period of the present study. As other studies have
demonstrated before, (e.g., van Pinxteren et al., 2019), the maps in Fig. 1 indicate the possibility
of imported pollution, in particular from the Southeast. Consequently, we searched for similar
trends in emission data reported by EEA for neighboring countries until 2017 directly West,
South, and East of Germany, going in the East all the way to Romania. Excel's solver optimized
combinations of the EEA-trends for Germany and neighboring countries in order to fit the
trends derived in the present study. The solver did not choose German trends for any of the
four parameters $PM_{10}$, BC, $SO_2$, and $NO_x$. For $PM_{10}$ a combination of emission trends for the
BENELUX countries and France was optimum, albeit without being able to simulate the
relative maxima in 2011 and 2013 and the minimum around 2016. For BC the emission trend
for the BENELUX countries came closest to the trend of the present study. For $SO_2$ mostly
emissions in Romania with minor contributions from French and BENELUX trends simulated
the trends observed over Germany best. $NO_x$-trends were best simulated by emissions over the
Czech and Slovakian countries. Emissions trends over Switzerland, Austria, Hungary and
Poland were not utilized by the solver. All simulated trends are displayed as curves EEA in
Fig. 6. We do not claim that these simulated trends numerically correspond to imported
pollution over Germany. However, the good fit of $SO_2$-trend with emissions over Romany
corroborates our finding of pollution import from Southeastern Europe to Northeastern
Germany while the development of BC appears to follow better emission trends over Western
neighbor countries than over Germany.

Sun et al., (2020) investigated trends of size resolved number and eBC mass concentrations
at 16 observational sites in Germany from 2009 to 2018 including the three GUAN-sites of the
present study. Based on monthly median time series they report average decreases for ME,
NG, and WA of -5.5%, -6.1, and -3.9%, respectively. The corresponding result for eBC of the
present study is -4.6%, albeit with a high variability (cf. Fig. 6) of 20 percent units expressed
in terms of a standard deviation.

Over the polluted continent the particle-number based parameter $N_{10-800}$ is largely secondary
in nature, i.e., its concentrations are controlled by atmospheric constituents and processes.
Thus, there is no primary emission data base with which a similar trend analysis as with $PM_{10}$,
BC, $SO_2$, and $NO_x$ could be attempted. Instead we chose the 10-year Grand Averages (GA)
averages taken over the whole time period of the present study as references from the deviations
of annual averages are discussed. Sun et al. (2020) report very minor trends (between -3.5%
and 0.1%) for $N_{20-800}$ at the three GUAN stations of the present study. The 10-year interannual
variation of our $N_{10-800}$ in Fig. 7a) bears out why only a minor trend if any can be expected. For
the first four years the annual averages are substantially higher than average. Then annual
values decrease down to a minimum in the years 2016/17 before they increase again to a level
slightly above the 10-year average.

In Figs 7b-d) annual deviations from the respective GAs are displayed that can be connected
to the 10-year course of $N_{10-800}$. Ozone concentrations averaged over the data from the three
GUAN stations can be interpreted as an indicator for photochemical activity that also controls
NPF. The annual deviations of $O_3$ in Fig. 7b) follow rather closely those of $N_{10-800}$. In Figs 7c)
and d) annual deviations of ambient temperature and precipitation rates are displayed that have
been averaged over the meteorological data along the back trajectories leading to the four
stations. For the temperature an averaging period of 120 trajectory hours yielded the highest
(negative) correlation with $N_{10-800}$ of r = -0.8. After a dip in 2009 annual average trajectory
temperatures to a maximum in 2016 before returning to near average in 2018. For the
precipitation rates along the trajectories the highest (negative) correlation with $N_{10-800}$ was
found with an averaging period of three days (r=-0.6) before arrival at the stations. The results
displayed in Figs 7c) and d) illustrate the complexity of processes and conditions controlling
atmospheric particle number concentrations. On one hand, a scavenging effect of precipitation
can be used as argument for the high values of $N_{10-800}$ in the years 2010-2013 and the low values
in the years 2014 through 2018. On the other hand, lower annual temperatures during years of
relatively high $N_{10-800}$ and higher than GA-temperatures during years of relatively high $N_{10-800}$
are harder to interpret. Possibly the nucleation of condensable vapors is furthered by lower air
temperatures upwind of the stations.

An important result of trend analysis are the average monthly factors disaggregating the
annual emissions. In general the summer minima of the month factors determined in the present
study are broader than the curve given by Matthias et al., (2018) for combustion emissions. The
decrease of the month factor of $PM_{10}$, BC, and $NO_x$ in December and the late winter maxima
of $PM_{10}$ and $SO_2$ are not reflected in the Matthias et al., (2018) results. Interestingly, both $PM_{10}$
and $SO_2$ show a minor secondary peak in June. As an example of the seasonal variability of
eBC within an urban source region we averaged the relative annual variation of eBC-
concentrations at the station Leipzig Eisenbahnstraße (plotted as curve L-EBS in Fig. 8)
exhibiting a smaller seasonal swing than all other curves. The curve for $PM_{10}$ comes closest to
that for L-EBS, probably because of agricultural non-combustion emissions in summer.

In general the downward trends in particulate parameters determined in the present study are
similar to temporal trends of particle number and black carbon mass concentrations at 16
observational sites in Germany from 2009 to 2018 (Sun et al., 2020). The long-term emission-
decrease of $PM_{10}$ as determined in the present study from 2009 to 2018 is smaller than the
corresponding number published by the EEA as average over all 28 EU member-states but
similar to the figures published by GEA until 2017 (cf. Table 2). For BC, $SO_2$, and $NO_x$ the
present study yields substantially stronger emission-reductions than both GEA and EEA. These
findings are emphasized when considering 2017 as endpoint of the trend calculation (cf. Table
2) at and after which our study shows consistent emission increases of all studied parameters.
Comparing the calculated trends with emission trends in neighboring countries as published by
the European Environment Agency supports the explanation that the observed trends are to
some extent due to changes in imported air masses. Most strongly this holds for $SO_2$, the trend
of which follows that of Romanian emissions rather well.

The last issue we take up in this discussion concerns the frequent residual difference between
measured and emission-simulated time series. In Fig. 5, e.g., in most winters there are months
when optimized BC-emissions remain substantially lower than the measured monthly medians
of eBC. Some information can be gleaned from the "Großwetterlagen", (GWL), representing
29 classifications of large scale weather types after Hess and Brezowsky for Central Europe,
(Gerstengarbe and Werner, 1993), provided by the German Weather service for each day
(http://www.dwd.de/DE/leistungen/grosswetterlage/grosswetterlage.html). During the winter
months with the strongest difference between measured and simulated time series the
probabilities of high-pressure systems over Fennoscandia with south-to-southeasterly flow to
the four stations is substantially higher than the respective probabilities averaged over the whole
ten-year period of the study. This GWL-information is consistent with the back trajectories
during the high pollution winter months coming predominantly from the southeasterly sector
of the map. While the classified large-scale weather situation with weak dilution of pollution
during the winter months is conducive of high particulate concentrations at the receptor sites it
does not explain the discrepancy. In principle our simplistic approach of accumulating
emissions along back trajectories may be flawed during certain weather situations. However,
an alternative explanation could be that the emissions inventories over Eastern and Southeastern
Europe in the EDGAR database are somewhat lower than the real emissions.


## 6 Summary and conclusions


Ten years of hourly aerosol and gas data at three stations of the German Ultrafine Aerosol
Network GUAN and one station of the Saxonian Environment Agency have been combined
with hourly back trajectories to the stations and emission inventories. Measured $PM_{10}$, particle
number concentrations between 10 and 800 nm, and equivalent black carbon were extrapolated
along the trajectories. This process yielded what we termed immission maps of these aerosol
parameters over Germany. They reflect aerosol emissions modified with atmospheric processes
along the air mass transport between sources and the four receptor sites at which potential
effects of the particulate air pollution would be realized.

The ten-year average immission maps do not simply show the distribution of pollution
sources upwind of the receptor sites. The comparison with emission data based on the European
EDGAR emission database shows that strong Western European emission centers do not
dominate the downwind concentrations because their emissions often are reduced by wet
scavenging and dilution processes on the way to the receptor area. Maps of average
precipitation and wind as they occurred along the trajectories illustrate these processes. In the
receptor region mass related aerosol parameters such as $PM_{10}$, equivalent black carbon, and to
some extent also the particle number concentration instead is rather controlled by emissions
from Eastern and Southeastern Europe from which pollution transport often occurs under dryer
meteorological conditions in continental high-pressure air masses. This finding corresponds to
the air mass results derived for the sub-micrometer particle number size distribution by Birmili
et al., (2001), by Engler et al., (2007) for the size distribution of non-volatile particles, by Ma
et al., (2014) for optical particle properties all evaluated at the station Melpitz, and by van
Pinxteren et al., (2019) for transboundary transport of $PM_{10}$ to ten stations in Eastern Germany
from neighboring countries. Newly formed particles on the other hand are found in air masses
from a broad belt reaching from Burgundy to the Western Czech Republic and Southern Poland,
a region with high photochemical activity in summer that is affected by emissions in Northern
Italy.

Annual EDGAR emissions for 2009 of $PM_{10}$, BC, $SO_2$, and $NO_x$, were accumulated along
each trajectory and compared the calculated emission sums with the corresponding measured
time series on a monthly basis. With a generalized reduced gradient solver the agreement of
each pair of monthly time series e.g., measured eBC and BC-emissions was optimized by letting
the solver determine both monthly emission factors disaggregating the annual EDGAR
emission fields and adjusting the emissions with annual factors modifying the 2009-fields.
Relative to 2009 the annual averages of the analyzed air pollutants were lower in 2018 by values
between 6% for $PM_{10}$ and 60% for $SO_2$. In general, the ten-year reductions determined of the
present study were stronger than those reported by the German and the European Environmental
Agencies. $N_{10\text{-}800}$ exhibited substantial interannual variability but no net decrease over the ten
studied years.

The validity of the present approach of connecting immission and emission of particulate
pollution was tested by calculating temporal changes of eBC for subsets of back trajectories
passing over two separate prominent emission regions, region A to the Northwest and B to the
Southeast of the measuring stations. Consistent with reported emission data the calculated
immission decreases over region A are significantly stronger than over region B.

Compared to published emission monthly factors by Matthias et al., (2018) the present
approach yielded broader summer minima that were partly displaced from the midsummer
positions given by Matthias et al., (2018). As an aside we note that during the winter months
with extremely high particulate pollution the emissions accumulated along back trajectories
often are substantially lower than the measured concentrations which raises the question of the
validity of the emission figures in Eastern and Southeastern European source regions.

There are clear limits in the methodology of the present study. Air mass trajectories have
inherent uncertainties increasing with their distance travelled (Stohl, 1998). Meteorological
processes affecting the aerosol during air mass transport are only considered rather coarsely
whereas aerosol dynamics are not considered at all. Possible future improvements concern
ensemble trajectories with higher resolution, better meteorological information along the
trajectories e.g., radar-derived precipitation as used in Heintzenberg et al., (2018), more
comprehensive emission inventories with higher spatiotemporal resolution and higher numbers
of analyzed stations.

Acknowledgements

This work was accomplished in the framework of the project ACTRIS-2 (Aerosols, Clouds,
and Trace gases Research InfraStructure) under the European Union—Research Infrastructure
Action in the frame of the H2020 program for "Integrating and opening existing national and
regional research infrastructures of European interest" under Grant Agreement N654109,
(H2020—Horizon 2020). Additionally, we acknowledge the WCCAP (World Calibration
Centre for Aerosol Physics) as part of the WMO-GAW program base-funded by the German
Federal Environmental Agency (UBA). Continuous aerosol measurements as well as data
processing at Melpitz, Waldhof and Neuglobsow were supported by the German Federal
Environment Agency Grants F&E 370343200 (German title: "Erfassung der Zahl feiner und
ultrafeiner Partikel in der Außenluft"), and F&E 371143232 (German title: "Trendanalysen
gesundheitsgefährdender Fein-und Ultrafeinstaubfraktionen unter Nutzung der im German
Ultrafine Aerosol Network (GUAN) ermittelten Immissionsdaten durch Fortführung und
Interpretation der Messreihen). We gratefully acknowledge receiving the emission data set
from European emission data base for global atmospheric research (EDGAR). We
acknowledge technical support by Annette Pausch of the Saxon State Office for Environment,
Agriculture and Geology at the Collmberg station, Achim Grüner und René Rabe (TROPOS)
at the Melpitz station, by Olaf Bath (GEA) at the Neuglobsow station, and Andreas Schwerin
(GEA) at the Waldhof station. Fabian Senf compiled the "Großwetterlagen" for the present
study. We are most grateful for the ideas provided by Peter Winkler in the interpretation of
data.

Literature
Anderson, J. O., Thundiyil, J. G., and Stolbach, A.: Clearing the air: a review of the effects of

particulate matter air pollution on human health, J Med Toxicol, 8, 166-175,

10.1007/s13181-011-0203-1, 2012.

Beekmann, M., Prévôt, A. S. H., Drewnick, F., Sciare, J., Pandis, S. N., Denier van der Gon,

H. A. C., Crippa, M., Freutel, F., Poulain, L., Ghersi, V., Rodriguez, E., Beirle, S.,

Zotter, P., von der Weiden-Reinmüller, S. L., Bressi, M., Fountoukis, C., Petetin, H.,

Szidat, S., Schneider, J., Rosso, A., El Haddad, I., Megaritis, A., Zhang, Q. J., Michoud,

695       V., Slowik, J. G., Moukhtar, S., Kolmonen, P., Stohl, A., Eckhardt, S., Borbon, A., Gros,

696       V., Marchand, N., Jaffrezo, J. L., Schwarzenboeck, A., Colomb, A., Wiedensohler, A.,

Borrmann, S., Lawrence, M., Baklanov, A., and Baltensperger, U.: In situ, satellite

measurement and model evidence on the dominant regional contribution to fine

particulate matter levels in the Paris megacity, Atmos. Chem. Phys., 15, 9577-9591,

10.5194/acp-15-9577-2015, 2015.

Birmili, W., Wiedensohler, A., Heintzenberg, J., and Lehmann, K.: Atmospheric particle

number size distribution in Central Europe: Statistical relations to air masses and

meteorology, J. Geophys. Res., 106, 32005-32018, 2001.

Birmili, W., Weinhold, K., Merkel, M., Rasch, F., Sonntag, A., Wiedensohler, A., Bastian, S.,

Schladitz, A., Löschau, G., Cyrys, J., Pitz, M., Gu, J., Kusch, T., Flentje, H., Quass, U.,

Kaminski, H., Kuhlbusch, T. A. J., Meinhardt, F., Schwerin, A., Bath, O., Ries, L.,

Wirtz, K., and Fiebig, M.: Long-term observations of tropospheric particle number size

distributions and equivalent black carbon mass concentrations in the German Ultrafine

Aerosol Network (GUAN), Earth Syst. Sci. Data, 8, 355-382, doi:10.5194/essd-8-355-

2016, 2016.

Bond, T. C., Doherty, S. J., Fahey, D. W., Forster, P. M., Berntsen, T., DeAngelo, B. J., Flanner,
M. G., Ghan, S., Kärcher, B., Koch, D., Kinne, S., Kondo, Y., Quinn, P. K., Sarofim,
M. C., Schultz, M. G., Schulz, M., Venkataraman, C., Zhang, H., Zhang, S., Bellouin,
N., Guttikunda, S. K., Hopke, P. K., Jacobson, M. Z., Kaiser, J. W., Klimont, Z.,
Lohmann, U., Schwarz, J. P., Shindell, D., Storelvmo, T., Warren, S. G., and Zender,
C. S.: Bounding the role of black carbon in the climate system: A scientific assessment,
J. Geophys. Res., doi: 10.1002/jgrd.50171, 10.1002/jgrd.50171, 2013.
Cass, G. R., and McRae, G. J.: Source-receptor reconciliation of routine air monitoring data for
trace metals: An emission inventory assisted approach, Environ. Sci. Technol., 17, 129-

720 139, 1983.

Charron, A., Birmili, W., and Harrison, R. M.: Fingerprinting particle origins according to their
size distribution at a UK rural site, J. Geophys. Res., 113, D07202,
doi:07210.01029/02007JD008562, 2008.
Crippa, M., Guizzardi, D., Muntean, M., Schaaf, E., Dentener, F., van Aardenne, J. A., Monni,
S., Doering, U., Olivier, J. G. J., Pagliari, V., and Janssens-Maenhout, G.: Gridded
emissions of air pollutants for the period 1970–2012 within EDGAR v4.3.2, Earth Syst.
Sci. Data, 10, 1987-2013, 10.5194/essd-10-1987-2018, 2018.
Diémoz, H., Barnaba, F., Magri, T., Pession, G., Dionisi, D., Pittavino, S., Tombolato, I. K. F.,
Campanelli, M., Della Ceca, L. S., Hervo, M., Di Liberto, L., Ferrero, L., and Gobbi, G.
P.: Transport of Po Valley aerosol pollution to the northwestern Alps – Part 1:
Phenomenology, Atmos. Chem. Phys., 19, 3065-3095, 10.5194/acp-19-3065-2019,
2019a.
Diémoz, H., Gobbi, G. P., Magri, T., Pession, G., Pittavino, S., Tombolato, I. K. F., Campanelli,
M., and Barnaba, F.: Transport of Po Valley aerosol pollution to the northwestern Alps
– Part 2: Long-term impact on air quality, Atmos. Chem. Phys., 19, 10129-10160,
10.5194/acp-19-10129-2019, 2019b.
Draxler, R., and Hess, G.: An overview of the HYSPLIT_4 modeling system for trajectories,
dispersion, and deposition, Austr. Meteor. Mag., 47, 295-308, 1998.

EEA: Spatial assessment of $PM_{10}$ and ozone concentrations in Europe (2005), European
Environmental Agency, Copenhagen, Denmark, 52 pp, 2009.

Eliassen, A.: The OECD Study of Long Range Transport of Air Pollutants: Long Range
Transport Modelling, Atmos. Environ., 12, 479-487, 1978.

Engler, C., Rose, D., Wehner, B., Wiedensohler, A., Brüggemann, E., Gnauk, T., Spindler, G.,
Tuch, T., and Birmili, W.: Size distributions of non-volatile particle residuals (Dp<800
745         nm) at a rural site in Germany and relation to air mass origin, Atmos. Chem. Phys., 7,
5785-5802, 10.5194/acp-7-5785-2007, 2007.

Friedlander, S. K.: Chemical element balances and identification of air pollution sources, Env.
Sci. & Technol., 7, 235-240, 10.1021/es60075a005, 1973.

Gerstengarbe, F.-W., and Werner, P. C.: Katalog der Grosswetterlagen Europas nach Paul Hess
und Helmut Brezowski 1881-1992, 4., vollständ. neu bearb. Aufl., Deutscher
Wetterdienst, Offenbach, Germany, 1993.

Heintzenberg, J., and Bussemer, M.: Development and application of a spectral light absorption
photometer for aerosol and hydrosol samples, J. Aerosol Sci., 31, 801-812, 2000.

Heintzenberg, J., Birmili, W., Seifert, P., Panov, A., Chi, X., and Andreae, M. O.: Mapping the
aerosol over Eurasia from the Zotino Tall Tower (ZOTTO), Tellus B, 65,
doi:http://dx.doi.org/10.3402/tellusb.v3465i3400.20062, 2013.

Heintzenberg, J., Leck, C., and Tunved, P.: Potential source regions and processes of aerosol
in the summer Arctic, Atmos. Chem. Phys., 15, 6487-6502, 10.5194/acp-15-6487-2015,
2015.

Heintzenberg, J., Senf, F., Birmili, W., and Wiedensohler, A.: Aerosol connections between
distant continental stations, Atmos. Environ., 190, 349-358, 2018.

Hůnová, I.: Ambient air quality for the territory of the Czech Republic in 1996–1999 expressed

by three essential factors, Sci. Total Environ., 303, 245-251,

https://doi.org/10.1016/S0048-9697(02)00493-X, 2003.

Hůnová, I., and Bäumelt, V.: Observation-based trends in ambient ozone in the Czech Republic

over the past two decades, Atmos. Environ., 172, 157-167,

https://doi.org/10.1016/j.atmosenv.2017.10.039, 2018.

Kanamitsu, M.: Description of the NMC Global Data Assimilation and Forecast System, Wea.

Forecasting, 4, 335-342, 10.1175/1520-0434(1989)004<0335:DOTNGD>2.0.CO;2,

1989.

Klimont, Z., Kupiainen, K., Heyes, C., Purohit, P., Cofala, J., Rafaj, P., Borken-Kleefeld, J.,

and Schöpp, W.: Global anthropogenic emissions of particulate matter including black

carbon, Atmos. Chem. Phys., 17, 8681-8723, 10.5194/acp-17-8681-2017, 2017.

Krige, D. G.: A statistical approach to some basic mine valuation problems on the

Witwatersrand, J. Chem. Metall. Min. Soc. S. Afr., December, 119-159, 1951.

Kuenen, J. J. P., Visschedijk, A. J. H., Jozwicka, M., and Denier van der Gon, H. A. C.: TNO-

MACC_II emission inventory; a multi-year (2003 - 2009) consistent high-resolution

European emission inventory for air quality modelling, Atmos. Chem. Phys., 14, 10963-

10976, 10.5194/acp-14-10963-2014, 2014.

Kulmala, M., Vehkamäkia, H., Petäjä, T., Dal Maso, M., Lauri, A., Kerminen, V.-M., Birmili,

781        W., and McMurry, P. H.: Formation and growth rates of ultrafine atmospheric particles:

a review of observations, J. Aerosol Sci., 35, 143-176, 2004.

Lasdon, L. S., Waren, A. D., Jain, A., and Ratner, M.: Design and Testing of a Generalized

Reduced Gradient Code for Nonlinear Programming, ACM Trans. Math. Softw., 4, 34–

50, 10.1145/355769.355773, 1978.

Lavanchy, V. M. H., Gäggeler, H. W., Schotterer, U., Schwikowski, M., and Baltensperger, U.:

Historical record of carbonaceous particle concentrations from a European high-alpine

glacier (Colle Gnifetti, Switzerland), J. Geophys. Res., 104, 21227-21236, 1999.

Leibensperger, E. M., Mickley, L. J., Jacob, D. J., Chen, W. T., Seinfeld, J. H., Nenes, A.,

Adams, P. J., Streets, D. G., Kumar, N., and Rind, D.: Climatic effects of 1950 - 2050

changes in US anthropogenic aerosols - Part 1: Aerosol trends and radiative forcing,

Atmos. Chem. Phys., 12, 3333-3348, 10.5194/acp-12-3333-2012, 2012.

Lelieveld, J., Evans, J. S., Fnais, M., Giannadaki, D., and Pozzer, A.: The contribution of

outdoor air pollution sources to premature mortality on a global scale, Nature, 525, 367-

371, 10.1038/nature15371, 2015.

Li, Z., Niu, F., Fan, J., Liu, Y., Rosenfeld, D., and Ding, Y.: Long-term impacts of aerosols on

the vertical development of clouds and precipitation, Nature Geosci., 4, 888-894, 2011.

Liu, S., Hua, S., Wang, K., Qiu, P., Liu, H., Wu, B., Shao, P., Liu, X., Wu, Y., Xue, Y., Hao,

Y., and Tian, H.: Spatial-temporal variation characteristics of air pollution in Henan of

China: Localized emission inventory, WRF/Chem simulations and potential source

contribution analysis, Sci. Total Environ., 624, 396-406,

https://doi.org/10.1016/j.scitotenv.2017.12.102, 2018.

Lugauer, M., and Winkler, P.: Thermal circulation in South Bavaria – climatology and synoptic

aspects, Meteor. Z., 14, 15-30, 2005.

Ma, N., Birmili, W., Müller, T., Tuch, T., Cheng, Y. F., Xu, W. Y., Zhao, C. S., and

Wiedensohler, A.: Tropospheric aerosol scattering and absorption over central Europe:

a closure study for the dry particle state, Atmos. Chem. Phys., 14, 6241-6259,

10.5194/acp-14-6241-2014, 2014.

Marmer, E., and Langmann, B.: Aerosol modeling over Europe: 1. Interannual variability of

aerosol distribution, J. Geophys. Res., 112, D23S15, doi:10.1029/2006JD008113, 2007.

Matthias, V., Arndt, J. A., Aulinger, A., Bieser, J., Denier van der Gon, H., Kranenburg, R.,

Kuenen, J., Neumann, D., Pouliot, G., and Quante, M.: Modeling emissions for three-

dimensional atmospheric chemistry transport models, Journal of the Air & Waste

Management Association, 68, 763-800, 10.1080/10962247.2018.1424057, 2018.

Miller, M. S., Friedlander, S. K., and Hidy, G. M.: A chemical element balance for the Pasadena

aerosol, J. Colloid Interface Sci., 39, 165-176, https://doi.org/10.1016/0021-

9797(72)90152-X, 1972.

Minkos, A., Dauert, U., Feigenspan, S., and Kessinger, S.: . German Environment Agency, Jan

2019, D-06813 , 28 pp. , Accessed on September 6, 2019 [Online] Available:

https://www.umweltbundesamt.de/sites/default/files/medien/1410/publikationen/1903

29_uba_hg_luftqualitaet_engl_bf.pdf: Air Quality 2018 - Preliminary Evaluation,

German Environment Agency, Dessau-Rosslau, Germany, 28, 2019.

Monks, P. S., Archibald, A. T., Colette, A., Cooper, O., Coyle, M., Derwent, R., Fowler, D.,

Granier, C., Law, K. S., Mills, G. E., Stevenson, D. S., Tarasova, O., Thouret, V., von

Schneidemesser, E., Sommariva, R., Wild, O., and Williams, M. L.: Tropospheric ozone

and its precursors from the urban to the global scale from air quality to short-lived

climate forcer, Atmos. Chem. Phys., 15, 8889-8973, 10.5194/acp-15-8889-2015, 2015.

Müller, T., Henzing, J. S., de Leeuw, G., Wiedensohler, A., Alastuey, A., Angelov, H., Bizjak,

829        M., Collaud Coen, M., Engström, J. E., Gruening, C., Hillamo, R., Hoffer, A., Imre, K.,

Ivanow, P., Jennings, G., Sun, J. Y., Kalivitis, N., Karlsson, H., Komppula, M., Laj, P.,

Li, S. M., Lunder, C., Marinoni, A., Martins dos Santos, S., Moerman, M., Nowak, A.,

Ogren, J. A., Petzold, A., Pichon, J. M., Rodriquez, S., Sharma, S., Sheridan, P. J.,

Teinilä, K., Tuch, T., Viana, M., Virkkula, A., Weingartner, E., Wilhelm, R., and Wang,

Y. Q.: Characterization and intercomparison of aerosol absorption photometers: result

of two intercomparison workshops, Atmos. Meas. Tech., 4, 245-268, 10.5194/amt-4-

245-2011, 2011.

Nordmann, S., Birmili, W., Weinhold, K., Müller, K., Spindler, G., and Wiedensohler, A.:
Measurements of the mass absorption cross section of atmospheric soot particles using
Raman spectroscopy, J. Geophys. Res., 118, 12,075-012,085, 10.1002/2013JD020021,

2013.

Panagos, P., Borrelli, P., Poesen, J., Ballabio, C., Lugato, E., Meusburger, K., Montanarella,
L., and Alewell, C.: The new assessment of soil loss by water erosion in Europe,
Environmental Science & Policy, 54, 438-447, 10.1016/j.envsci.2015.08.012, 2015.
Patashnick, H., and Rupprecht, E. G.: Continuous PM-10 Measurements Using the Tapered
Element Oscillating Microbalance, Journal of the Air & Waste Management
Association, 41, 1079-1083, 10.1080/10473289.1991.10466903, 1991.
Penner, J. E., Dong, X., and Chen, Y.: Observational evidence of a change in radiative forcing
due to the indirect aerosol effect, Nature, 427, 231-234, 2004.
Petzold, A., Ogren, J. A., Fiebig, M., Laj, P., Li, S. M., Baltensperger, U., Holzer-Popp, T.,
Kinne, S., Pappalardo, G., Sugimoto, N., Wehrli, C., Wiedensohler, A., and Zhang, X.
Y.: Recommendations for reporting "black carbon" measurements, Atmos. Chem.
Phys., 13, 8365-8379, 10.5194/acp-13-8365-2013, 2013.
Pruppacher, H. R., and Klett, J. D.: Microphysics of Clouds and Precipitation, Reidel Publishing
Co., Dordrecht, 714pp, 1978.
Rehme, R.: Application of Gas Phase Titration in the Calibration of Nitric Oxide, Nitrogen
Dioxide, and Ozone Analyzers, in: Calibration in Air Monitoring, edited by: Chapman,
R., and Sheesley, D., ASTM International, West Conshohocken, PA, 198-209, 1976.
Reitebuch, O., Dabas, A., Delville, P., Drobinsk, P., and Gantner, L.: Characterization of Alpine
pumping by airborne Doppler lidar and numerical simulations., Int. Conf. Alp. Meteor.,
Brig 2003. – Publications of MeteoSwiss, 66, 134-137, 2003.
Riemer, N., Vogel, H., and Vogel, B.: Soot aging time scales in polluted regions during day
and night, Atmos. Chem. Phys., 4, 1885-1893, 2004.
Rohde, R. A., and Muller, R. A.: Air Pollution in China: Mapping of Concentrations and

Sources, PLoS One, 10, e0135749-e0135749, 10.1371/journal.pone.0135749, 2015.

Samset, B. H., Sand, M., Smith, C. J., Bauer, S. E., Forster, P. M., Fuglestvedt, J. S., Osprey,

S., and Schleussner, C. F.: Climate Impacts From a Removal of Anthropogenic Aerosol

Emissions, Geophysical Research Letters, 45, 1020-1029, 10.1002/2017gl076079,

2018.

Schell, B., Ackermann, I., Hass, H., Binkowski, F., and Ebel, A.: Modeling the formation of

secondary organic aerosol within a comprehensive air quality model system, J.

Geophys. Res., 106, 28275–28293, 2001.

Schwartz, S. E.: The whitehouse effect - shortwave radiative forcing of climate by

anthropogenic aerosols: an overview, J. Aerosol Sci., 27, 359-382, 1996.

Sen, P. K.: Estimates of the Regression Coefficient Based on Kendall's Tau, J. Am. Stat.

Assoc., 63, 1379–1389, 1968.

Spindler, G., Müller, K., and Herrmann, H.: Main particulate matter components in Saxony

(Germany) - trends and sampling aspects, Environ. Sci. Pollut. Res., 6, 89-94, 1999.

Spindler, G., Grüner, A., Müller, K., Schlimper, S., and Herrmann, H.: Long-term size-

segregated particle (PM10, PM2.5, PM1) characterization study at Melpitz -- influence

of air mass inflow, weather conditions and season, J. Atmos. Chem., 70, 165-195,

10.1007/s10874-013-9263-8, 2013.

Stein, A. F., Draxler, R. R., Rolph, G. D., Stunder, B. J. B., Cohen, M. D., and Ngan, F.:

NOAA's HYSPLIT Atmospheric Transport and Dispersion Modeling System, Bull.

Amer. Meteor. Soc., 96, 2059-2077, 10.1175/BAMS-D-14-00110.1, 2015.

Stohl, A.: Trajectory statistics - a new method to establish source-receptor relationships of air

pollutants and its application to the transport of particulate sulfate in Europe, Atmos.

Environ., 30, 579-587, 1996.

Stohl, A.: Computations, accuracy and applications of trajectories - A review and bibliography,

Atmos. Environ., 32, 947-966, 1998.

Struzewska, J., and Jefimow, M.: A 15-year analysis of surface ozone pollution in the context

of hot spells episodes over Poland, Acta Geophysica, 64, 1875-1902, 10.1515/acgeo-

2016-0067, 2013.

Sun, J., Birmili, W., Hermann, M., Tuch, T., Weinhold, K., Spindler, G., Schladitz, A., Bastian,

S., Löschau, G., Cyrys, J., Gu, J., Flentje, H., Briel, B., Asbach, C., Kaminski, H., Ries,

895        L., Sohmer, R., Gerwig, H., Wirtz, K., Meinhardt, F., Schwerin, A., Bath, O., Ma, N.,

and Wiedensohler, A.: Variability of Black Carbon Mass Concentrations, Sub-

micrometer Particle Number Concentrations and Size Distributions: Results of the

German Ultrafine Aerosol Network Ranging from City Street to High Alpine Locations,

Atmos. Environ., 202, 256-268, https://doi.org/10.1016/j.atmosenv.2018.12.029, 2019.

Sun, J., Birmili, W., Hermann, M., Tuch, T., Weinhold, K., Merkel, M., Rasch, F., Müller, T.,

Schladitz, A., Bastian, S., Löschau, G., Cyrys, J., Gu, J., Flentje, H., Briel, B., Asbach,

C., Kaminski, H., Ries, L., Sohmer, R., Gerwig, H., Wirtz, K., Meinhardt, F., Schwerin,

903        A., Bath, O., Ma, N., and Wiedensohler, A.: Decreasing Trends of Particle Number and

Black Carbon Mass Concentrations at 16 Observational Sites in Germany from 2009 to

2018, Atmos. Chem. Phys., 2019, 1-19, 10.5194/acp-2019-754, 2020.

Swietlicki, E., Svantesson, B., and Hansson, H.-C.: European source area apportionment, J.

Aerosol Sci., 19, 1175-1178, 1988.

Theil, H.: A Rank-Invariant Method of Linear and Polynomial Regression Analysis, in: Henri

Theil's Contributions to Economics and Econometrics: Econometric Theory and

Methodology, edited by: Raj, B., and Koerts, J., Springer Netherlands, Dordrecht, 345–

381, 1992.

Twomey, S.: Pollution and the planetary albedo, Atmos. Environ., 8, 1251-1256, 1974.
van Pinxteren, D., Mothes, F., Spindler, G., Fomba, K. W., and Herrmann, H.: Trans-boundary

PM10: Quantifying impact and sources during winter 2016/17 in eastern Germany,

Atmos. Environ., 200, 119-130, https://doi.org/10.1016/j.atmosenv.2018.11.061, 2019.

Wehner, B., Siebert, H., Stratmann, F., Tuch, T., Wiedensohler, A., Petäjä, T., Dal Maso, M.,

and Kulmala, M.: Horizontal homogeneity and vertical extent of new particle formation

events, Tellus, 59 B, 362-371, 2007.

Wichmann, H. E., and Peters, A.: Epidemiological evidence of the effects of ultrafine particle

exposure, Philosophical Transactions of the Royal Society of London, 358, 1751-2769,

2000.

Wiedensohler, A., Birmili, W., Nowak, A., Sonntag, A., Weinhold, K., Merkel, M., Wehner,

B., Tuch, T., Pfeifer, S., Fiebig, M., Fjäraa, A. M., Asmi, E., Sellegri, K., Depuy, R.,

Venzac, H., Villani, P., Laj, P., Aalto, P., Ogren, J. A., Swietlicki, E., Williams, P.,

Roldin, P., Quincey, P., Hüglin, C., Fierz-Schmidhauser, R., Gysel, M., Weingartner,

E., Riccobono, F., Santos, S., Grüning, C., Faloon, K., Beddows, D., Harrison, R.,

Monahan, C., Jennings, S. G., O'Dowd, C. D., Marinoni, A., Horn, H. G., Keck, L.,

Jiang, J., Scheckman, J., McMurry, P. H., Deng, Z., Zhao, C. S., Moerman, M., Henzing,

B., de Leeuw, G., Löschau, G., and Bastian, S.: Mobility particle size spectrometers:

harmonization of technical standards and data structure to facilitate high quality long-

term observations of atmospheric particle number size distributions, Atmos. Meas.

Tech., 5, 657-685, 10.5194/amt-5-657-2012, 2012.

Wiedensohler, A., Wiesner, A., Weinhold, K., Birmili, W., Hermann, M., Merkel, M., Müller,

934        T., Pfeifer, S., Schmidt, A., Tuch, T., Velarde, F., Quincey, P., Seeger, S., and Nowak,

935        A.: Mobility particle size spectrometers: Calibration procedures and measurement

uncertainties, Aerosol Sci. Technol., 52, 146-164, 10.1080/02786826.2017.1387229,

2018.

Winkler, P., Lugauer, M., and Reitebuch, O.: Alpine Pumping, PROMET, 32, 34-42, 2006.

Wolke, R., Hellmuth, O., Knoth, O., Schröder, W., Heinrich, B., and Renner, E.: The chemistry-

transport modeling system LM-MUSCAT: Description and CityDelta applications, in:

Air Pollution Modeling and its Application XVI, Kluwer Academic/Plenum, 427–439,

2004.

Zanatta, M., Gysel, M., Bukowiecki, N., Müller, T., Weingartner, E., Areskoug, H., Fiebig, M.,

Yttri, K. E., Mihalopoulos, N., Kouvarakis, G., Beddows, D., Harrison, R. M., Cavalli,

F., Putaud, J. P., Spindler, G., Wiedensohler, A., Alastuey, A., Pandolfi, M., Sellegri,

946        K., Swietlicki, E., Jaffrezo, J. L., Baltensperger, U., and Laj, P.: A European aerosol

phenomenology-5: Climatology of black carbon optical properties at 9 regional

background sites across Europe, Atmos. Environ., 145, 346-364,

10.1016/j.atmosenv.2016.09.035, 2016.

Zhu, K., Zhang, J., and Lioy, P. J.: Evaluation and Comparison of Continuous Fine Particulate

Matter Monitors for Measurement of Ambient Aerosols, Journal of the Air & Waste

Management Association, 57, 1499-1506, 10.3155/1047-3289.57.12.1499, 2007.


Table 1: Characteristics of the four stations of the present study, see text for instrumental details.  The number of validated data hours are

given for each component

| Station | Acronym | Latitude | Longitude | MPSS[1] | eBC[2] | PM10 continous[3,4] | PM10 discontinous[5] | NOx[6] | SO2[7] | O3[8] |
|---|---|---|---|---|---|---|---|---|---|---|
| Collmberg | CO | 51.3 | 13 | | | 85054 | | 88838 | | 88792 |
| Melpitz | ME | 51.5 | 12.9 | 81561 | 88196 | | 88822 | 86260 | 85541 | 84421 |
| Neuglobsow | NG | 53.1 | 13 | 57962 | 77540 | 71202 | | 83718 | 87778 | 87943 |
| Waldhof | WA | 52.8 | 10.8 | 84276 | 80725 | 88321 | | 85503 | 82386 | 87373 |

[1]MPSS - scanning mobility particle size spectrometer TROPOS $(10-800$ nm); [2]MAAP - Multi-angle absorption photometer 5012 Thermo Fischer Scientific; [3]TEOM-FDM - Tapered element oscillating microbalance fitted with a filter dynamics measuring system 1405 Thermo Fischer Scientific; [4]SCHARP - Synchronized Hybrid Ambient Real-time Particulate Monitor 5030 Thermo Fischer Scientific; [5]HVS – High Volume Sampler DIGITEL DH-80; [6]TLA-NOx –Trace Level NOx Analyzer 42i-TL Thermo Fischer Scientific; [7]TLA-SO2 - Trace Level $SO_2$ Analyzer 43i-TLE Thermo Fischer Scientific; [8]



Table 2 Median concentrations of eBC concentrations ($\mu gm^{-3}$) and temporal trends (2009-2018) of eBC in terms of Sen-Theil slope (% per year) as determined for air masses passing over Regions A and B as analyzed at the stations Melpitz (ME), Neuglobsow (NG), and Waldhof (WA). For comparison the national annual decreases in BC emissions 2009-2017 in % according to the European Environmental Agency are added.

| | DELTA T* in h | No. of back trajectories | | | Median eBC in μm/m³ | | | Sen-Theil slope in % per year | | | | Decrease in national BC emissions in % per year | | |
|---|---|---|---|---|---|---|---|---|---|---|---|---|---|---|
| | | ME | NG | WA | ME | NG | WA | ME | NG | WA | 3 Stations** | Belgium | Netherlands | Germany |
| | 1 | 21941 | 17514 | 27218 | 0.38 | 0.40 | 0.41 | -6.40 | -6.80 | -4.80 | -5.85 | -6.1% | -6.1% | -4.9% |
| Region A | 3 | 18605 | 14268 | 22132 | 0.38 | 0.40 | 0.41 | -6.40 | -6.90 | -4.80 | -5.89 | | | |
| B-NL-NRW | 6 | 14802 | 10086 | 15936 | 0.39 | 0.40 | 0.42 | -6.40 | -7.60 | -5.10 | -6.19 | | | |
| | 12 | 6817 | 3746 | 6131 | 0.40 | 0.50 | 0.50 | -7.10 | -7.90 | -5.30 | -6.62 | | | |
| | | | | | | | | | | | | Czech Rep. | Poland | Slovakia |
| | 1 | 11096 | 5264 | 4191 | 1.10 | 1.19 | 1.13 | -3.60 | -3.40 | -1.70 | -3.16 | -2.8% | 0.5% | -2.3% |
| Region B | 3 | 9601 | 4339 | 3541 | 1.08 | 1.18 | 1.12 | -3.40 | -3.40 | -2.10 | -3.14 | | | |
| CZ-PL-SK | 6 | 7000 | 3062 | 2570 | 1.05 | 1.09 | 1.11 | -4.00 | -2.90 | -2.70 | -3.47 | | | |
| | 12 | 3628 | 1410 | 1277 | 1.00 | 1.00 | 1.00 | -3.70 | -3.00 | -2.70 | -3.34 | | | |
| ALL | | 85846 | 75190 | 78356 | 0.45 | 0.36 | 0.36 | -5.90 | -5.60 | -4.00 | -5.18 | | | |
| Sun (2020) | | | | | | | | -4.40 | -7.80 | -3.20 | | | | |

* Minimum time spent over the specified source region, **Weighted mean, according to the available number of back trajectories

Table 3    Percental decreases in the anthropogenic emissions of $PM_{10}$, BC, $SO_2$, and

$NO_x$ relative to 2009 as reported by the European Environment Agency (EEA,

https://www.eea.europa.eu/data-and-maps/dashboards/air-pollutant-emissions-data-

viewer-2), the German Environment Agency (GEA), and calculated in the present

study.  The EEA and GEA only report data until 2017, (*=BC until 2016).


| Component | EEA 2009-2017 | GEA 2009-2017 | GUAN emissions 2009-2017 | GUAN emissions 2009-2018 |
|---|---|---|---|---|
| $PM_{10}$ | 12% | 4.2% | 16% | 6% |
| BC* | 29% | 35%* | 63% | 44% |
| $SO_2$ | 33% | 20% | 68% | 59% |
| $NO_x$ | 20% | 11% | 43% | 30% |



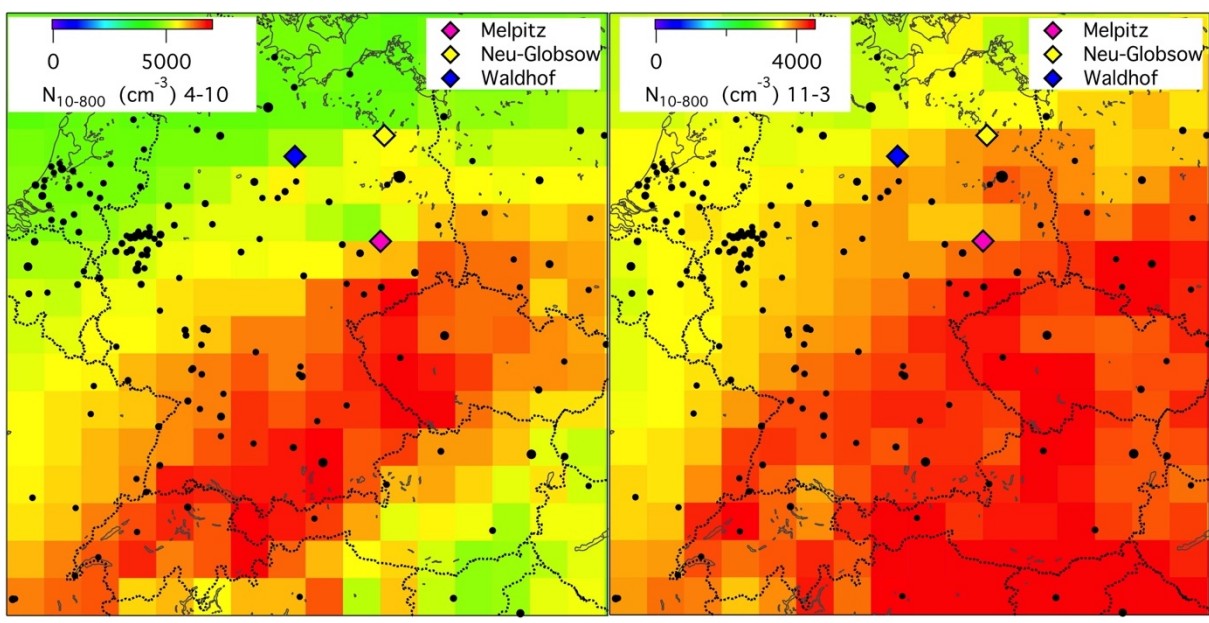
Fig. 1   Maps of particle number concentration $N_{10-800}$ $(cm^{-3})$ extrapolated under 1000 m height

along five day back trajectories from hourly data at the four stations from 2009 to 2018;

left: months April through October; right: months November through March. The

GUAN-stations are marked with colored diamonds. The Collmberg station lies 30 km

Southeast of station Melpitz. Here and in the following maps the black dots represent

cities larger than 100000 inhabitants with the size of the dots being proportional to the

number of inhabitants.



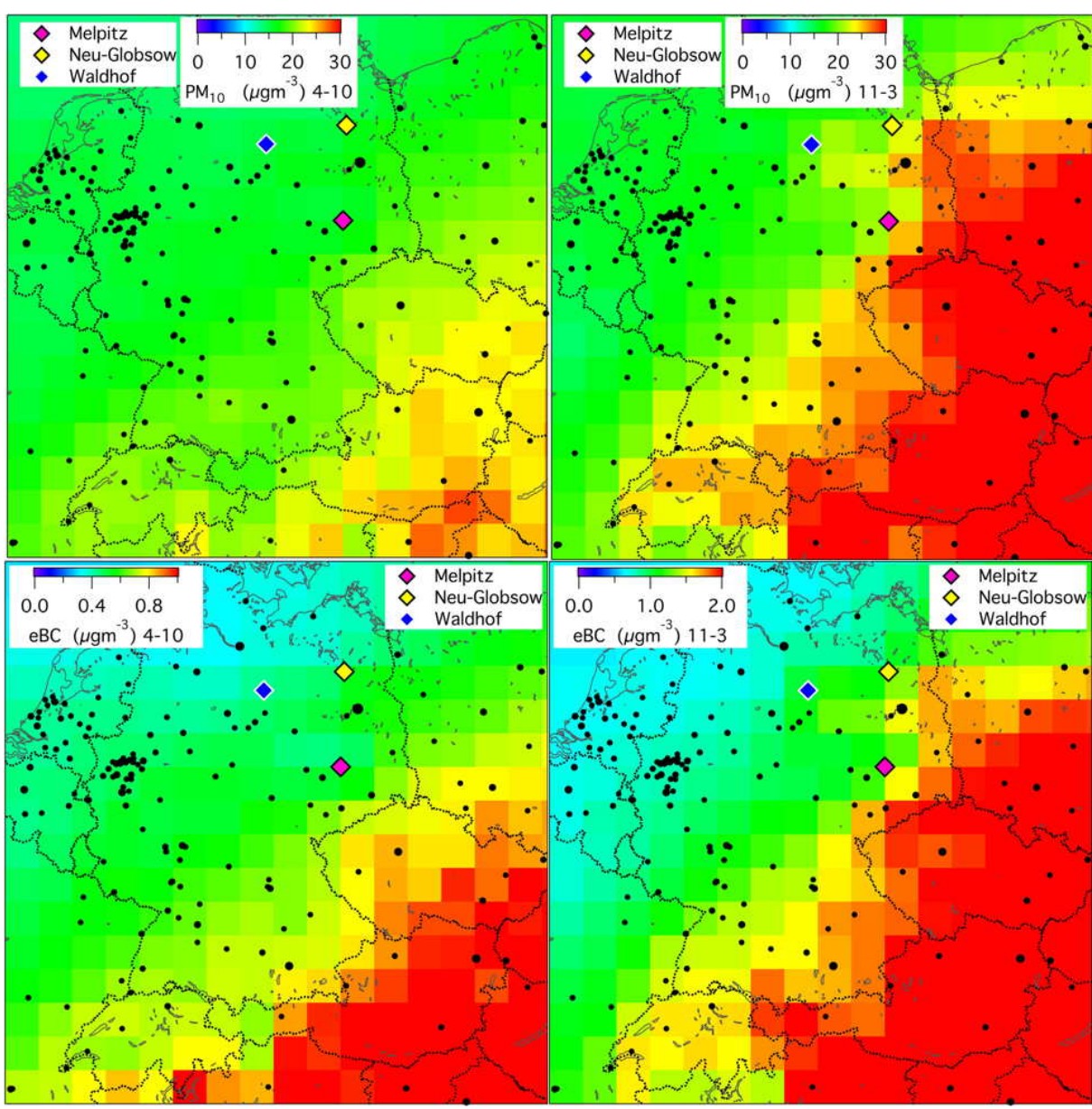

Fig. 2  As Fig. 1 but for particle mass concentrations (top, PM₁₀, µgcm⁻³), and black carbon

concentrations (bottom, eBC, µgm⁻³).


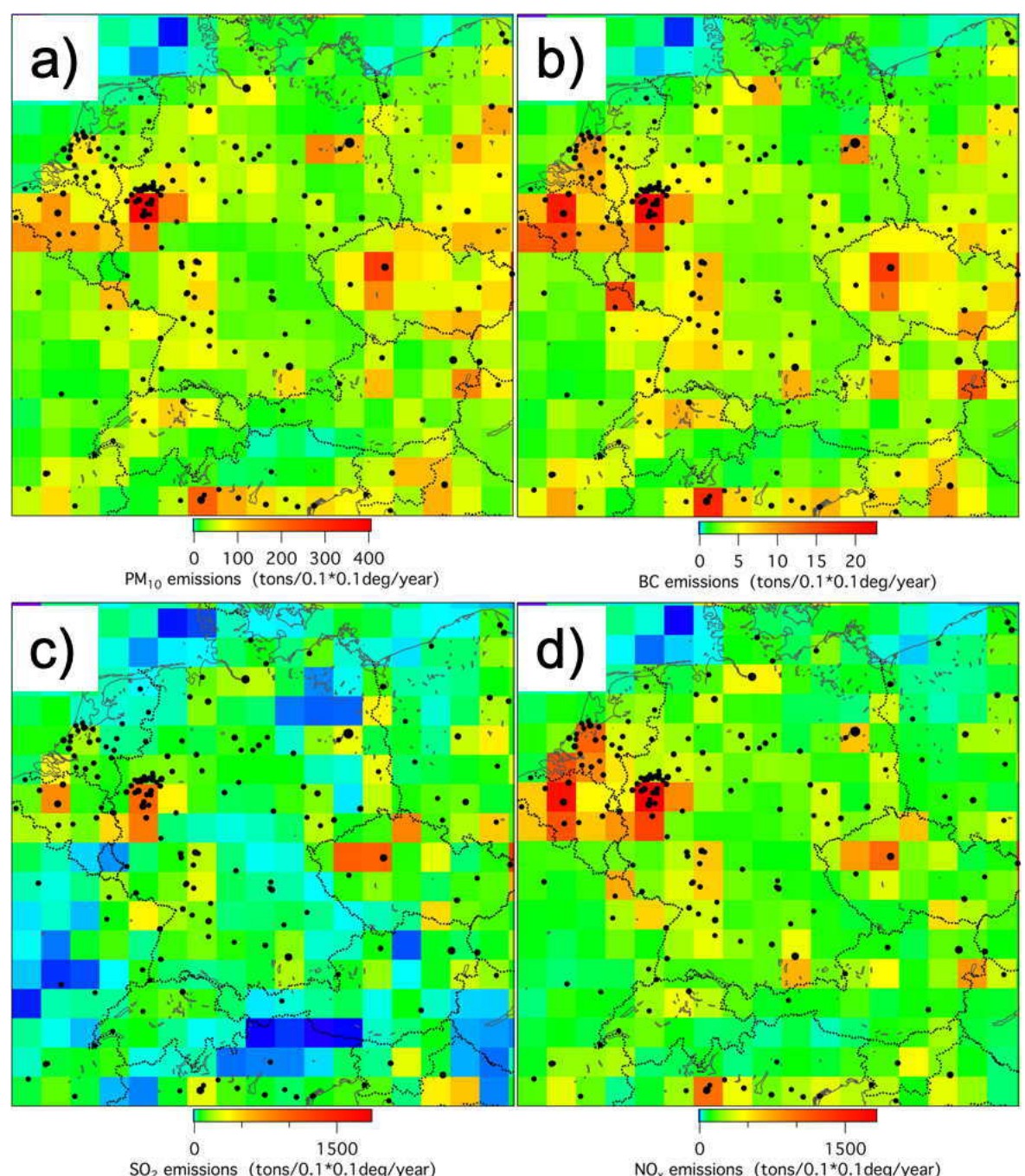


Fig. 3 As Fig. 1 but a) for PM₁₀ emissions (tons/0.1*0.1deg./year), b) for BC emissions, c) for

SO₂ emissions, and d) for NOₓ emissions (tons/0.1*0.1deg./year) according to the

EDGAR                                    emission                                    database

(https://data.europa.eu/doi/10.2904/JRC_DATASET_EDGAR) for 2009 averaged over

the geogrid of the present study.


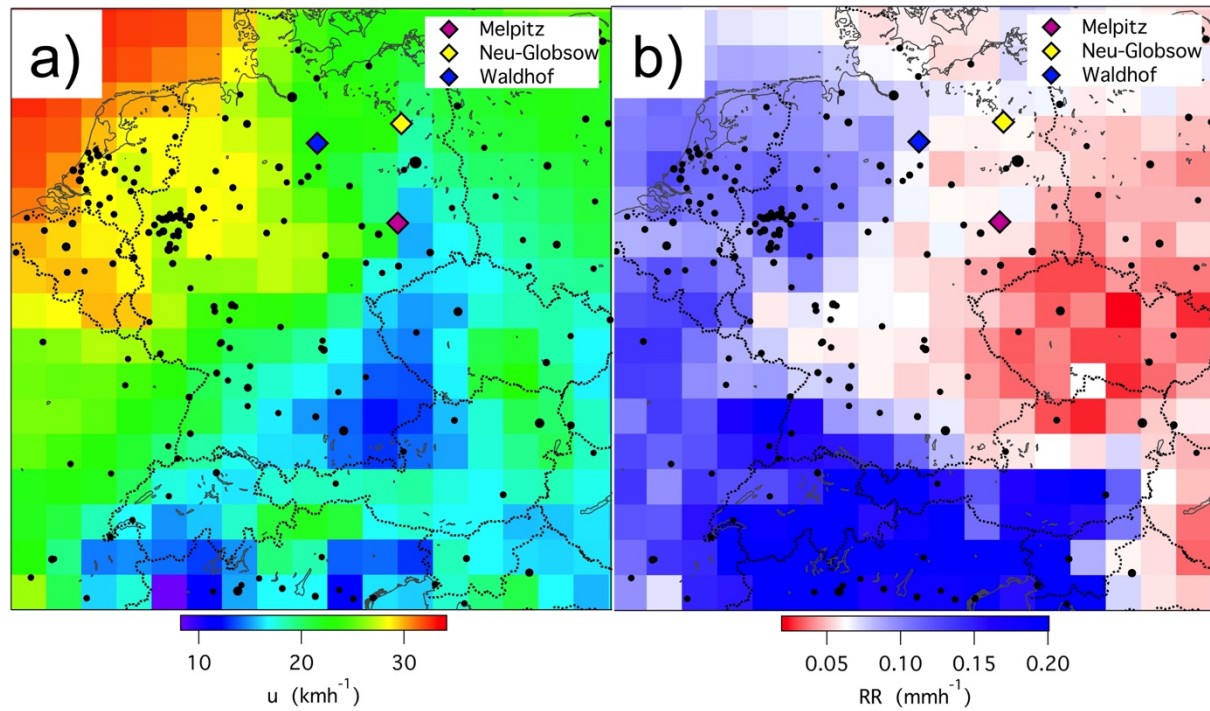


Fig. 4  a) Map of horizontal wind speed (u, kmh$^{-1}$) as reported by HYSPLIT along hourly five-

992          day back trajectories to the four stations marked in the graph averaged over the time

period 2009 to 2018; b) as a) but for precipitation (RR, mmh$^{-1}$).


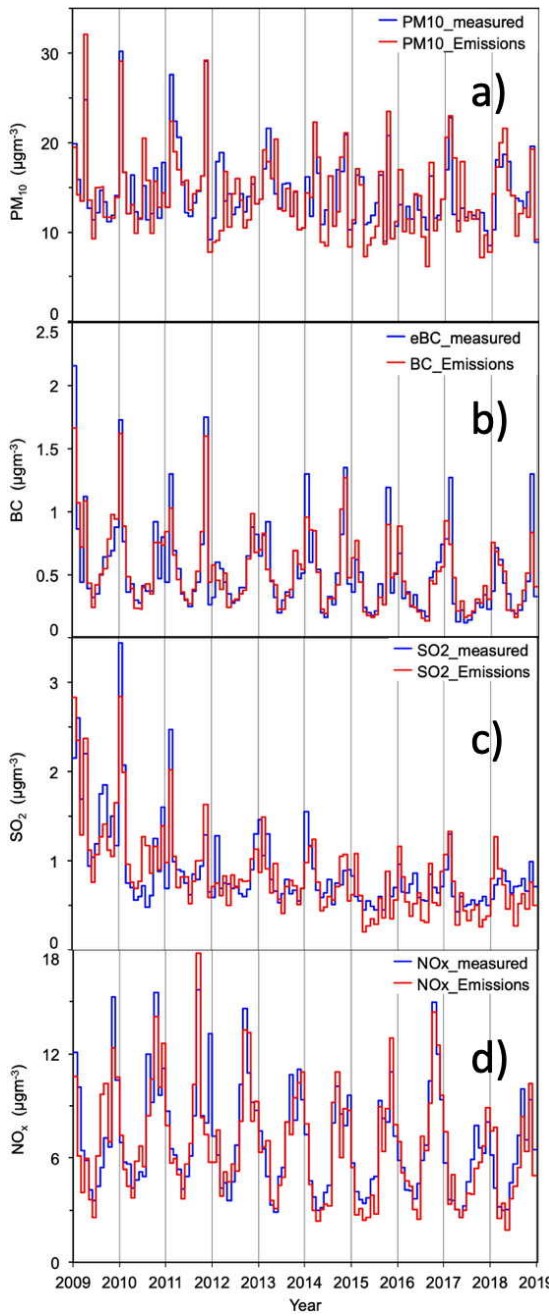


Fig. 5  a) Monthly medians of $PM_{10}$-concentrations at the four stations of the present study

(blue), and monthly medians of optimized sums of $PM_{10}$-emissions along back

trajectories leading to the stations (red). b) as a) but for measured eBC-concentrations

and BC-emissions along back trajectories. c) as a) but for measured $SO_2$-concentrations

and $SO_2$-emissions along back trajectories. d) as a) but for measured $NO_x$-

concentrations and $NO_x$ -emissions along back trajectories.

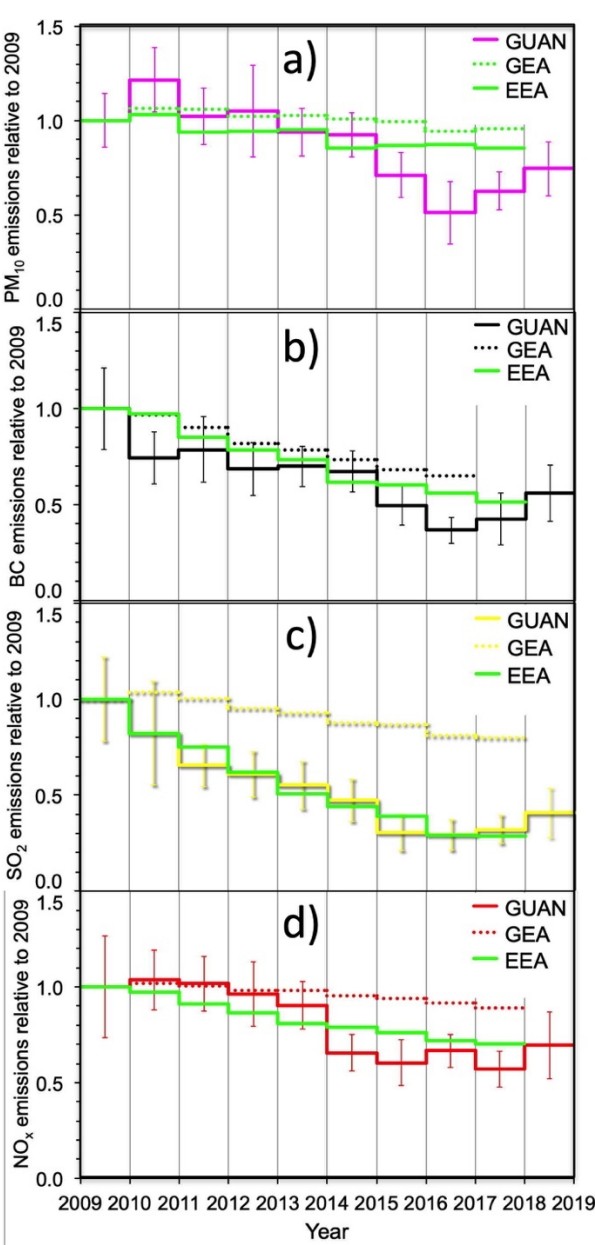

Fig. 6  GUAN: Trends in the emissions of a) $PM_{10}$, b) BC, c) $SO_2$, and d) $NO_x$, relative to 2009

as calculated by optimizing the agreement between 2009-EDGAR-emissions and

concentrations measured at the four stations of the present study. The error bars

represent annual average relative deviations between measured and simulated data.

GEA: Trends as reported for Germany by the German Environment Agency. EEA:

Trends as optimized from combinations of trends over Germany and neighboring

countries, (see text for details).


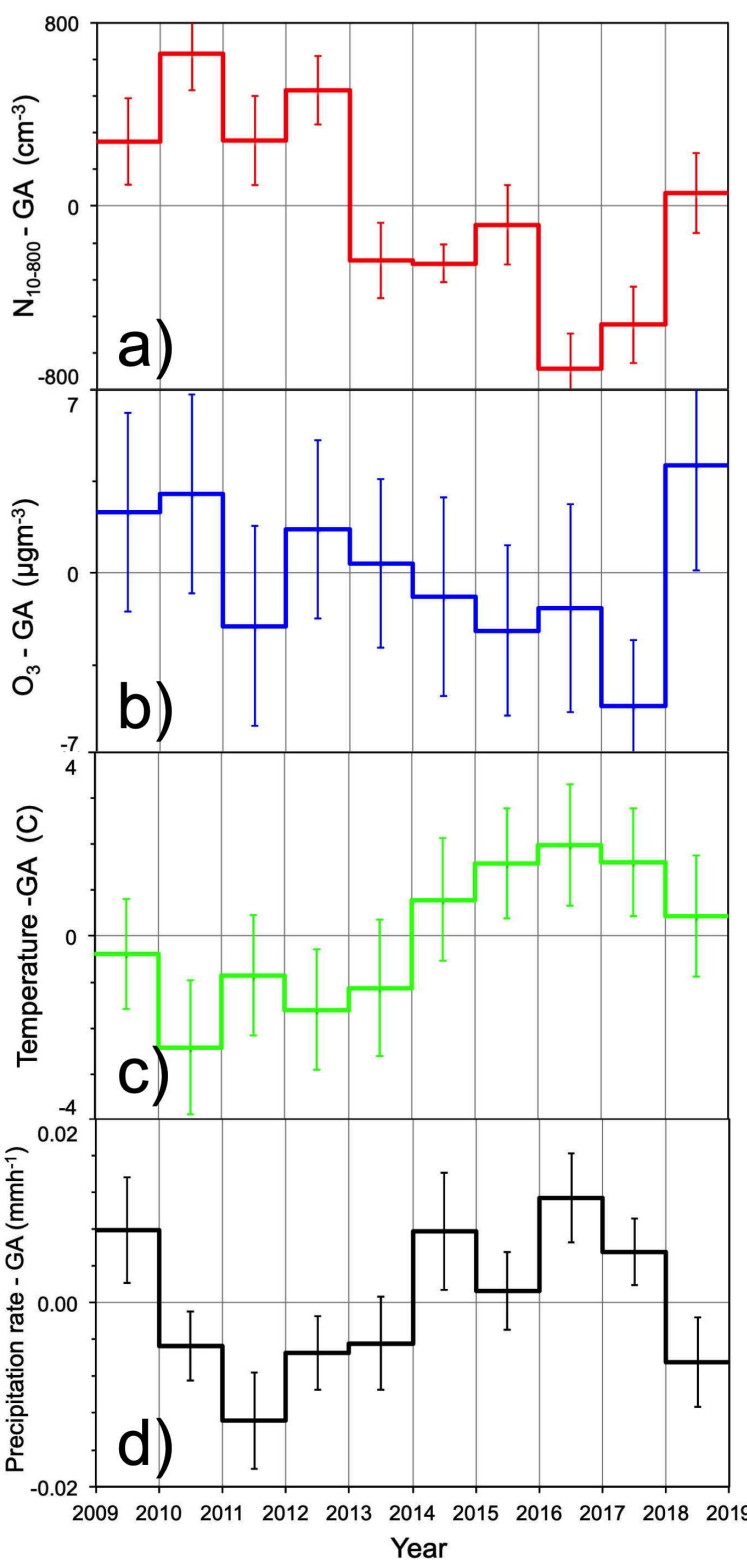


Fig. 7 Trends in annual average deviations a) $\Delta N_{10\text{-}800}$, b) $\Delta O_3$, c) temperature $\Delta T$ along the

trajectories five days back in time, and d) precipitation rate $\Delta RR$ along the trajectories

three days back in time. The deviations are taken relative to the respective 10-year

Grand Average (GA).  The error bars represent the standard deviations of the annual
averages.

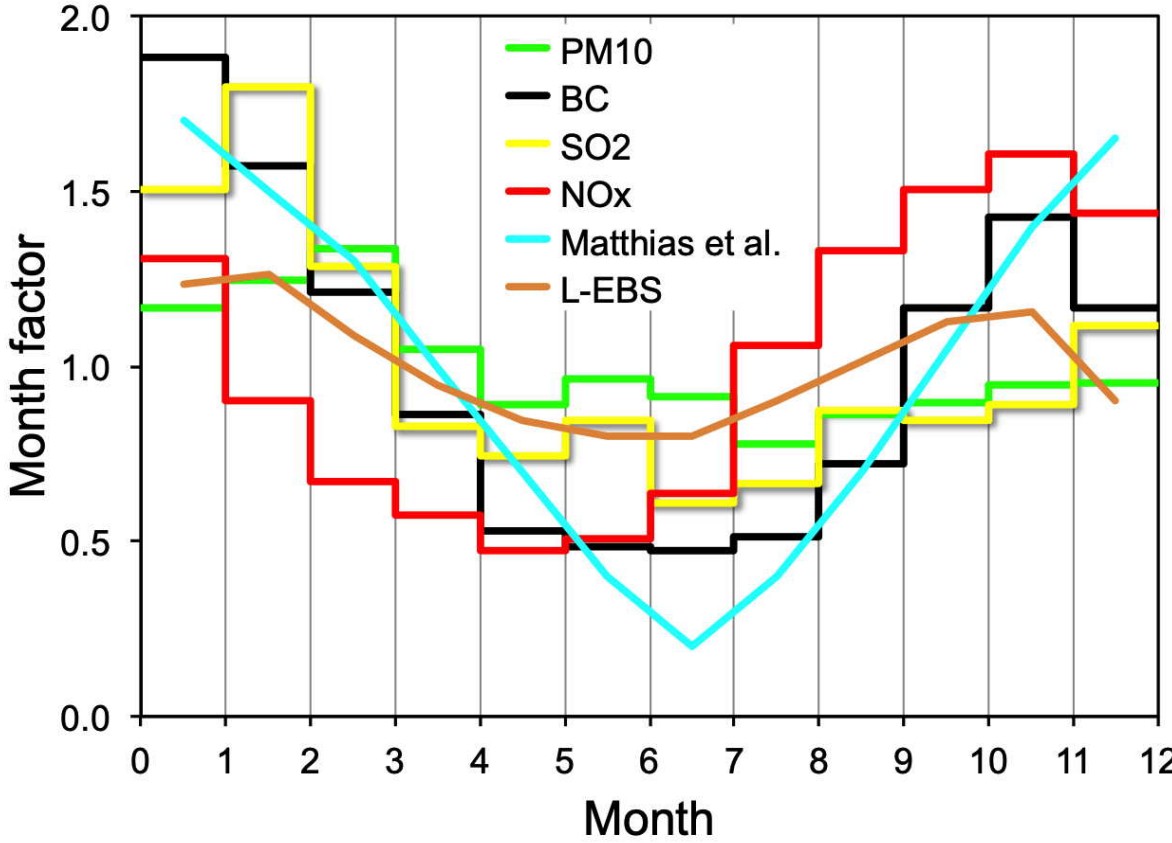

Fig. 8 Month factors for the emissions of $PM_{10}$, BC, $SO_2$, and $NO_x$ as determined by
optimizing the agreement between EDGAR-emissions and concentrations measured at
the four stations of the present study. For comparison the month factors of Matthias et
al., (2018) for combustion emissions are plotted and the relative annual variation of eBC
concentrations measured at the station Leipzig-Eisenbahnstraße (L-EBS) averaged over
the time period of the present study.
