# Peer review of "Aerosol immission maps and trends over Germany with hourly data at four rural background stations from 2009 to 2018 3 4 Jost Heintzenberg1, Wolfram Birmili2, Bryan Hellack2, Gerald Spindler1, Thomas Tuch1, and Alfred Wiedensohler1 5 6 1: Leibniz Institute for Tropospheric Research (TROPOS), Permoserstr. 15, 04318 Leipzig, 7 Germany 8 2: German Environment Agency, Wörlitzer Platz 1, 06844 Dessau-Ro"

_Atmospheric Chemistry and Physics, 2019_

## Referee Comment (RC1) · Anonymous Referee #1 · 27 Mar 2020

In this paper, hourly measurements of aerosol and gases at four rural sites in Germany during ten years have been analyzed by combining the observations with back trajectories. The goal was to identi-fy potential source regions by connecting back trajectories and emission inventories. The back trajectory approach was chosen because chemical transport models are difficult to use for such long time periods. The paper is certainly of interest to the readers of ACP. However, the paper is in my view lacking de-tailed information about the applied data analytical methods making it very difficult or even impossible to understand and rate the conclusions of this study. In my view, this manuscript requires major revi-sions before it can be published in ACP (see main comments below).

[Figure]

Main comments:

The methodical part of the back trajectory analysis is explained in section 3. There is, however, no de-scription how exactly the back trajectories are used and how the maps are calculated. This methodical chapter must be much more detailed (and might also include the most important equations) so that the reader can reconstruct what exactly has been done. This is necessary for understanding the meaning, limitations and the interpretation of the maps as given by the authors later in the paper. Some examples: It remains unclear how the authors treated back trajectories that did not interact with the planetary boundary layer. How exactly were source regions defined? On page 8, line 172 the authors state that "Precipitation along the trajectories was used in the interpretation of the immission maps". How has this been done? Without this information it is not possible for the reader to evaluate the quality of this work and the relevance of the findings. Also for the second approach (Page 11, lines 259/261), there is no information how exactly the emissions were summed up. I expect that emissions were only summed up when the air parcel was close to the surface, but the reader doenn't know. The applied approach must be explained in more detail.

On page 10 the authors write that the calculated immission maps have little in common with the emis-sion data as shown in Fig. 2. Later in lines 239 and 240 it is said that "dilution through mixing with cleaner air masses and wet scavenging through in-cloud and sub-cloud processes" are two major at-mospheric processes during transport. This is certainly true, it remains however unclear how such processes are incorporated in the back trajectory analysis. As the paper is now, I have the impression that the calculated immission maps are largely dominated by such atmospheric processes and the interpretation in terms of potential source regions is difficult. The signal from the potential source regions seem to be masked by such processes, e.g. strong dilution of emissions during westerly winds.

For the second approach (connection of emissions and aerosol observations), an op-timization algo-rithm was applied. There is again hardly any information what exactly

has been done, except that the name of the used solver is given. This should be changed, please explain the applied optimization method in more detail. Maybe provide the relevant equations.

Page 7, lines 139-141: The authors say that equivalent back carbon was measured using a MAAP and converting the measured absorption signal into black carbon by using the default mass absorption cross section of the instrument (6.6 m2g-1). For using the terminology of equivalent black carbon (eBC) it is to my knowledge required that the mass absorption cross section is determined specifically for the site of interest, e.g. from parallel measurements with elemental carbon (EC) as determined using an analytical method such as the thermal optical method (see cited paper by Petzold et al 2013). The authors should therefore not call their measurements equivalent black carbon (eBC) but instead black carbon (BC).

Page 13, lines 298 – 300. The authors write that contributions from wind erosion of agricultural soils are not incorporated in present anthropogenic inventories. What about anthropogenic precursor emissions of secondary PM? How are they included or not included in this study. The authors should provide the corresponding discussion and information.

Page 28, Table 2. It is surprising that the calculated percentage change in PM10 emissions relative to 2009 is highly sensitive to the considered time period (for 2009 – 2017 percentage change is -16%, for 2009 – 2018 percentage change is -6%). This deserves some explanation: Can this be a matter of limited robustness of the applied method, what are the uncertainties of the trend estimations?

Minor comments:

Page 7, line 143 – 144: At three sites TEOM1400 instruments were used for PM10 measurement, at one of the sites daily PM10 was gravimetrically determined. The authors should say a few words about the comparability of the two methods as it is known that there can be (and typically are) systematic differences.

Page 7, lines 157-158: "Through combustion processes the trace gases NOx and SO2 are related to anthropogenic aerosol formation". This is not correct or at least unclear. Of course, NOx and SO2 are in Germany mainly emitted from anthropogenic sources and processed in the atmosphere to form aerosols. However, the sentence should be re-phrased to something like "NOx and SO2 emitted by anthropogenic combustion processes are transformed in the atmosphere and add to the anthropogenic aerosol".

Page 9, line 204: What is the "Southeastern half of the map"? Please be more accurate. Page 9, line 207: Typo, "PM10and". Page 9, line 212: It is unclear what countries/region is meant here? Rephrase so that this is clear Page 12, line 279: Delete "are". Page 12, line 288: I cannot see an increase in PM10 in 2010, neither in Fig. 4 nor in Fig. 5. Please explain or revise.

---

## Referee Comment (RC2) · Anonymous Referee #2 · 30 Mar 2020

This manuscript investigates aerosol and trace gas concentrations and emission changes in northern Europe based on a combination of atmospheric measurements, emission inventory and air mass back trajectory calculations. The investigation has a potential to be scientifically interesting but, in its present form, lacks information and details that should be incorporated. My main comments in this regard are given below.

The background information for this study presented in section 1 is written very well. However, the text is totally European centered. For completeness for a reader not that familiar with this topic, it would nice to have at least one paragraph shortly summarizing whether similar work has been done outside Europe, for example in Norther America

or Asia.

There are a few issues related the applied data. First, the paper does not report anything about data coverage or its quality. Were there any major data gaps, not apparent in Table 1, that might influence the analysis performed in the paper? Is all the data quality checked and how? Do the detection limits of different instruments, especially what it comes to trace gas data, play any role here? Second, there are no continuous PM10 data for Melpitz, neither SO2 data for Collmberg. How does this influence the analysis of this paper? Third, what is purpose of presenting PM0.8 derived from SMPS if that is not used anywhere in the paper? One final issue: Figure 1 caption refers to particle volume concentration when showing PM10. I suppose this is a mistake.

What is the benefit of displaying particle number concentrations plots (Figs. 1, top and bottom left) for the rest of this paper? Particle number concentrations are not related to any of the discussions about emissions later in the paper. The authors briefly mention atmosphere new particle formation as one particle source, and emphasizes SO2 plumes or lofted layers as important locations for this source. If these two sources dominated new particle formation over Europe, how would they explain the common observation of regional new particle formation events, taking simultaneously place over tens to hundreds of km scales, in a vast number of surface measurement sites in Europe.

As mentioned by the authors, PM10 in Figs. 1 and 2 have practically no resemblance with each other. Noting that both these data sources are used together in later analyses, the authors should discuss this issue and its potential consequences in their analysis. What are the potential reasons for the differences between these two figures? Why the map in Figure 1 does not capture the emission hotspots of Figure 2? There is strong PM10 gradient from North-West to South-East in Figure 1. Why does Figure 2 not give any indication about such gradient? Are the main PM10 in the South-East direction located outside the map area?

The authors state that NOx an SO2 maps look very similar (related to figures 1 and 2). I wonder why these results are shown in the paper, especially when considering that both these trace gases are important parts of trend analyses presented later in the paper.

Based on Figure 3, the authors discuss the influence of wind speed and wet scavenging on measured concentrations. Unfortunately, this discussion remains very qualitative. If included in this paper, more concrete results on the influence of these two variables should be presented.

In the last part of the paper, the authors investigate the time evolution of emissions during the past 10 years. I have a couple of concerns related to this. First, the authors only stated that they optimized the annual and monthly emissions of 2009 for the rest of the time series using their measurements. This is too scarce information. The authors should provide more details on how this exercise what done in practice. Second, as well known, emission changes have been very different in different parts of Europe during the past years. Is the approach applied in this paper able to catch this feature in any way, and if it is not, this should be explicitly mentioned in the paper.

The final conclusion of this paper remain somewhat vague. The authors should more concrete summarize what new information this study brings on top of what is already known about the past emission changes in Europe. The authors mention some of the limitations of the current study yet, based on the comments give above, I feel that this list could be expanded a bit.

---

## Author Response (AR1)

**Rev. 1**

**Comment:**

The methodical part of the back trajectory analysis is explained in section 3. There is, however, no description how exactly the back trajectories are used and how the maps are calculated. This methodical chapter must be much more detailed (and might also include the most important equations) so that the reader can reconstruct what exactly has been done. This is necessary for understanding the meaning, limitations and the interpretation of the maps as given by the authors later in the paper. Some examples: It remains unclear how the authors treated back trajectories that did not interact with the planetary boundary layer. How exactly were source regions defined? Response: The revised text has been greatly expanded in response to the criticism by both reviewers. We refer to specific answers to the corresponding questions by rev. 2.

**Comment:**

On page 10 the authors write that the calculated immission maps have little in common with the emission data as shown in Fig. 2. Later in lines 239 and 240 it is said that "dilution through mixing with cleaner air masses and wet scavenging through in-cloud and sub-cloud processes" are two major atmospheric processes during transport. This is certainly true, it remains however unclear how such processes are incorporated in the back trajectory analysis. As the paper is now, I have the impression that the calculated immission maps are largely dominated by such atmospheric processes and the interpretation in terms of potential source regions is difficult. The signal from the potential source regions seem to be masked by such processes, e.g. strong dilution of emissions during westerly winds.

Response: This comment rightly touches one of the key points of our paper, namely our results that illustrate that emission maps not necessarily can be equated with the distribution of effect-related **immissions** in given receptor areas.

To clarify the information about HYPLIT4 we added to the model introduction: "Turbulent atmospheric mixing is included in parameterized form in HYSPLIT4. The present study utilizes the default version of this parameterization according to Draxler and Hess (1998). The back trajectories are calculated with the base version of HYSPLIT4 that does not include any specific dispersion and scavenging of atmospheric trace substances."

**Comment:**

For the second approach (connection of emissions and aerosol observations), an optimization algorithm was applied. There is again hardly any information what exactly has been done, except that the name of the used solver is given. This should be changed, please explain the applied optimization method in more detail. Maybe provide the relevant equations. Response:

We greatly enlarged the respective text, including two equations:

. We expected both, seasonal variations and a long-term trend in the emissions. For M hours

per month of measured components at the four stations the annual average EDGAR-emissions

 $E_{PMIO}$ ,  $E_{BC}$ ,  $E_{SO2}$ , and  $E_{NOx}$  were summed up along the 121 trajectory steps leading to the stations.

Then monthly medians  $\tilde{E}_{i=1,4}$  were formed according to Eq. 1 (exemplified for BC). Medians

were chosen to reduce the effect of outliers due to local emission and scavenging events.

$$\tilde{E}_{BC} = Median(\sum_{n=1}^{121} E_{BC})_{m=1,M}$$
Eq. 1

The monthly median emission sums  $\tilde{E}_{i=1,4}$  were modified with a monthly  $(f_{m})$  and an annual factor  $(g_{n})$  in order to simulate respective median monthly measured concentrations taken over all stations. Thus, for each component 12 monthly and 10 annual trend factors determined the agreement of modified summed emissions and measured concentrations. As objective or utility function  $\chi^{2}$  the sum of squared deviations between annually and monthly modified emission sums and monthly median measured concentrations was formed taken over the 120 months of the present study (exemplified for BC in Eq. 2).

$$\chi_{BC}^{2} = \sum_{j=1}^{120} (f_{m=1,12} \cdot g_{y=1,10} \cdot \tilde{E}_{BC} - eBC)^{2}$$
 Eq. 2

 $\chi^2$  was minimized with a Generalized Reduced Gradient (GRG) solver (Lasdon et al., 1978) that optimized the12 monthly and 10 annual factors for each of the four measured components. We used Excel's® implementation of the GRG solver procedure for the optimization. After optimization of month and trend factors the average relative deviation between emission-simulated and measured monthly median curves is 14%, 21%, 25%, and 18% for PM10, eBC, SO2, and NOx, and respectively

**Comment:**

Page 7, lines 139-141: The authors say that equivalent back carbon was measured using a MAAP and converting the measured absorption signal into black carbon by using the default mass absorption cross section of the instrument (6.6 m2g-1). For using the terminology of equivalent black carbon (eBC) it is to my knowledge required that the mass absorption cross section is determined specifically for the site of interest, e.g. from parallel measurements with elemental carbon (EC) as determined using an analytical method such as the thermal optical method (see cited paper by Petzold et al 2013). The authors should therefore not call their measurements equivalent black carbon (eBC) but instead black carbon (BC) Response:

**In the instrumental section we added the two references (Birmili et al., 2016; Nordmann et al., 2013) that give details on the ways that the MAAP data at the GUAN-stations are validated against two analytical methods (Raman spectroscopy and thermal analysis) as requested by the reviewer. Consequently, we would like to maintain our use of term eBC.**

Comment:

On page 8, line 172 the authors state that "Precipitation along the trajectories was used in the interpretation of the immission maps". How has this been done? Response:

An explanation was given in the next sentence of the original manuscript, obviously not enough. Thus, we enlarged the respective text to:

Precipitation along the trajectories was used in the interpretation of the immission maps. The precipitation values mapped in the present study are those listed by HYSPLIT4 at each point of a trajectory. They are precipitation rates at the nearest grid cell of the assimilated global meteorological fields provided by the U.S. National Weather Service's National Centers for Environmental Prediction (NCEP) (Kanamitsu, 1989).

**Comment:**

Page 13, lines 298 – 300. The authors write that contributions from wind erosion of agricultural soils are not incorporated in present anthropogenic inventories. What about anthropogenic precursor emissions of secondary PM? How are they included or not included in this study. The authors should provide the corresponding discussion and information.

Response: The EDGAR data base utilized in the present study covers primary emissions only. Thus, our discussion is limited accordingly. In the revised text of the introduction of the EDGAR data base this has been clarified.

**Comment:**

Page 28, Table 2. It is surprising that the calculated percentage change in PM10 emissions relative to 2009 is highly sensitive to the considered time period (for 2009 – 2017 percentage change is -16%, for 2009 – 2018 percentage change is -6%). This deserves some explanation: Can this be a matter of limited robustness of the applied method, what are the uncertainties of the trend estimations?

Response: This change in percentage change is due to the general increase in emissions after 2017 that is clearly seen in Fig. 5 and in recent preliminary data shown by the European Environmental Agency (e.g., https://www.eea.europa.eu/data-and-maps/dashboards/air-quality-statistics-expert-viewer). Concerning the uncertainties of the trend estimations we added error bars to the trend figures. As estimates of the uncertainties these error bars were calculated by averaging the 12 monthly relative differences between measured and emission-simulated measurements for each year and applying these annual average relative differences as relative uncertainties to the calculated trends.

**Comment:**

Page 7, lines 157-158: "Through combustion processes the trace gases NOx and SO2 are related to anthropogenic aerosol formation". This is not correct or at least unclear. Of course, NOx and SO2 are in Germany mainly emitted from anthropogenic sources and processed in the atmosphere to form aerosols. However, the sentence should be re-phrased to something like "NOx and SO2 emitted by anthropogenic combustion processes are transformed in the atmosphere and add to the anthropogenic aerosol". Response: Text changed as requested.

**Comment:**

*Page 9, line 204: What is the "Southeastern half of the map"? Please be more accurate.* Response: Sorry, but we do not really understand the reviewer's problem here. The complete incriminated sentence reads "Highest average concentrations are measured in 204 airmasses from the Southeastern half of the map, most strongly expressed in PM10 and eBC 205 with maxima in a region leading from Southwest Poland through the Czech Republic, Slovakia, 206 Austria, and former Yugoslavia to Northeastern Italy, "i.e., the countries meant by "Southeastern half of the map" are specified.

**Comment:**

Page 9, line 207: Typo, "PM10and". Response: Typo corrected.

Comment:

Page 9, line 212: It is unclear what countries/ region is meant here? Rephrase so that this is clear,

Response: The incriminated part of the sentence reads: "...the emissions in the former countries stayed nearly constant...". We used the term "former" because we did not want to burden the reader by repeating the list of countries specified 3 sentences before but repeated it now in the revised text: "the emissions in Poland, Czech Republic, Slovakia, Austria, former Yugoslavia, and Italy stayed nearly constant or even increased in recent years"

Comment:

Page 12, line 279: Delete "are". Response: Done

Comment:

*Page 12, line 288: I cannot see an increase in PM10 in 2010, neither in Fig. 4 nor in Fig. 5. Please explain or revise.*

Response: The original text was unclear. We revised it to: "In 2010/2011 the trend curves of PM10 and NOx in Fig. 5 show a slight increase that can be linked to a recovery of economic activity after the world-wide financial and economic crisis during the period 2007-2009. The  $\mathbf{PM}_{10}$ the relative increase in is also visible in trend curves to 2005 published by the German Environment Agency

(https://www.umweltbundesamt.de/daten/luft/luftschadstoff-emissionen-in-deutschland/emissionen-prioritaerer-luftschadstoffe).

**Rev. 2**

**Comment:**

For completeness for a reader not that familiar with this topic, it would nice to have at least one paragraph shortly summarizing whether similar work has been done outside Europe, for example in Norther America or Asia.

Comment: Outside Europe we only succeeded to identify two comparable studies in China for which we added the text: "With a much larger data set spanning a much tighter network of 1500 stations in China Rohde and Muller (2015) used the Kriging interpolation approach (Krige, 1951) to construct air pollution maps over China. Another approach to construct pollution maps over the province Henan, China was used by Liu et al., (2018). They combined an emission inventory with chemical modeling and back trajectories to derive high resolution maps of particulate and gaseous pollution components."

**Comment:**

First, the paper does not report anything about data coverage or its quality. Were there any major data gaps, not apparent in Table 1, that might influence the analysis performed in the paper?

Response: Table 1 has been augmented with the number of validated data hours for each measured component and the text related to Table 1 has been extended to: "Table 1 gives an overview over the instrumental characteristics of all stations and the total number of validated data hours for each utilized component. The minimum is 57962 hours for validated SMPS-data at the tree GUAN-stations and the maximum with 88838 validated data hours for NO4 at all four stations. Strictly concurrent (by the hour) are less validated data hours. For SMPS, eBC, and SO2-data at the GUAN-stations this numbers is 48533 hours, and 48114 and 47729 hours for PM10 and NO4-data, respectively, at all four stations. However, these reduced strictly concurrent numbers do not substantially affect the 10-year-average maps discussed below."

**Comment:**

**Is all the data quality checked and how? Do the detection limits of different instruments, especially what it comes to trace gas data, play any role here?**

Response: The text on data quality has been greatly expanded and reads now as "TROPOStype mobility particle size spectrometers (MPSS, Wiedensohler et al., 2012) were used to record particle number size distributions across the particle size range 10-800 nm. Quality assurance of the long-term measurements followed the recommendations of Wiedensohler et al. (2018) including weekly inspections as well as monthly and annual maintenance intervals. Once a year the MPSS were intercompared against a reference MPSS of the WCCAP (World Calibration Center for Aerosol Physics) either on-site and/or at the calibration facility. The lower detection limit of the MPSS is around 100 cm-3 for a time resolution of 5 minutes. Equivalent Black Carbon (eBC) was determined by multi-angle absorption photometers (MAAP) using a mass absorption cross section of 6.6 m2 g-1 (Petzold et al., 2013; Nordmann et al., 2009; Birmili et al., 2009). An intercomparison of multiple MAAP instruments resulted in an inter-device variability of less than 5% (Müller et al., 2011). While the MAAP deployed at the TROPOS station Melpitz was biannually compared to the reference absorption photometer at the WCCAP in Leipzig, the instruments at the UBA stations Waldhof and Neuglobsow were serviced by the manufacturer. For hourly measurements of PM10 continuous oscillating microbalances (Thermo Scientific TEOM 1400) were utilized at stations CO, NG, and WA. At station ME PM10 data were determined in daily filter samples (0:00 to 24:00 CET), Spindler et al. (2013). The TEOM1400-instrument and gravimetric filter sampling are different methods for particle mass concentrations. The TEOM collects particulate mass on a vibrating substrate (tapered element) and registers the change of the oscillation frequency that is decreasing with mass loading (Patashnick and Rupprecht, 1991). The TEOM operates at a constant temperature setting above ambient (typically 30- 50°C) to prevent contraction and expansion of the tapered element and reduce interferences from water vapor condensation. However, heating the ambient air enhances volatilization of particle-bound semivolatile compounds (e.g., ammonium nitrate and some organic species) resulting in an underestimation of PM when semivolatile dominate the particulate phase during cold seasons. The condensation and evaporation of ammonium nitrate and organic species can also influence the filter sampling under ambient conditions. Here the effect can be balanced partly by the temperature variation during the daily filter sampling. However, the results of both methods mostly are in good agreement (e.g., Zhu et al., 2007)....At the three GUAN stations both are measured with the same temporal resolutions as the aerosol data. Additionally, at Collmberg NOx-data could be utilized in the interpretation of the aerosol data. The trace gas analyzers for NOx and SO2 were calibrated with test gases for NO (NO in N2) and SO2 (SO2 in N2, both Air Liquide, Germany). NO2 was produced in a gas phase titration device (GPT APMC370, Horiba, Germany) by quantitative oxidation of NO test gas (Rehme, 1976). The trace gas analyzers were used in an optimal range and all registered values (also below the detection limit) were used for this longterm study.

**Comment:**

Second, there are no continuous PM10 data for Melpitz, neither SO2 data for Collmberg. How does this influence the analysis of this paper?

Response: Basically, eliminating one of the stations from the maps only reduces the statistical solidity in some of the geocells but not the overall geographic patterns. In particular the continuous PM10-data at Collmberg located only 30 km away from Melpitz supports the daily average PM10-data at Melpitz. We added the following text and two references stating that TEOM1400 and gravimetric filter sampling are different methods but are mostly in good agreement.

"The TEOM1400-instrument and gravimetric filter sampling are different methods for particle mass concentrations. The TEOM collects particulate mass on a vibrating substrate (tapered element) and registers the change of the oscillation frequency that is decreasing with mass loading (Patashnick and Rupprecht, 1991). The TEOM operates at a constant temperature setting above ambient (typically 30– 50°C) to prevent contraction and expansion of the tapered element and reduce interferences from water vapor condensation. However, heating the ambient air enhances volatilization of particle-bound semivolatile compounds (e.g., ammonium nitrate and some organic species) resulting in an underestimation of PM when semivolatile dominate the particulate phase during cold seasons. The condensation and evaporation of ammonium nitrate and organic species can also influence the filter sampling under ambient conditions. Here the effect can be balanced partly by the temperature variation during the daily filter sampling. However, the results of both methods mostly are in good agreement (e.g., Zhu et al., 2007)."

For the measurements described in this paper there are also results from a successful one-year intercomparison of TEOM1400 and filter sampling in winter 2012/13 in the region (city of Dresden) of special interest. Results available at https://www.luft.sachsen.de/download/luft/Vergleichsmessungen PM2 5 Internet.pdf

The highest time resolution discussed in the present study is one month while most results are discussed as 10-year averages. Thus, utilizing daily  $PM_{10}$ -averages from the Melpitz station did not affect the results significantly.

**Comment:**

Third, what is purpose of presenting PM0.8 derived from SMPS if that is not used anywhere in the paper?

Response: Sorry about that. This was just a typo that had not been eliminated from a previous version of the manuscript.

**Comment:**

One final issue: Figure 1 caption refers to particle volume concentration when showing PM10. Again, sorry about that. This was just a typo that had not been eliminated from a previous version of the manuscript.

**Comment:**

What is the benefit of displaying particle number concentrations plots (Figs. 1, top and bottom left) for the rest of this paper? Particle number concentrations are not related to any of the discussions about emissions later in the paper. The authors briefly mention atmosphere new particle formation as one particle source, and emphasizes SO2 plumes or lofted layers as important locations for this source. If these two sources dominated new particle formation over Europe, how would they explain the common observation of regional new particle formation events, taking simultaneously place over tens to hundreds of km scales, in a vast number of surface measurement sites in Europe.

Response: Thank you for criticizing the lack of motivation and limited discussion of particlenumber related results. First, we eliminated the discussion of  $N_{10.26}$  because of the more locally controlled nature of this parameter. Then we expanded the discussion of  $N_{10.80}$  substantially with seasonal maps and a trend discussion connected to gas phase and meteorological parameters.

**Comment:**

As mentioned by the authors, PM10 in Figs. 1 and 2 have practically no resemblance with each other. Noting that both these data sources are used together in later analyses, the authors should discuss this issue and its potential consequences in their analysis. What are the potential reasons for the differences between these two figures? Why the map in Figure 1 does not capture the emission hotspots of Figure 2? There is strong PM10 gradient from North-West to South-East in Figure 1. Why does Figure 2 not give any indication about such gradient? Are the main PM10 in the South-East direction located outside the map area?

Response: In the section on emissions we first qualified the emission data by "Primary aerosol emission data are generally characterized by rather high uncertainties. For the EDGAR data base Crippa et al. (2018) report a range of variation in 2012 between 57.4% and 109.1% for PM10, and between 46.8% and 92% for BC. Even higher uncertainties in PM emissions might come from super-emitting vehicles that are not considered in this data base (Klimont et al., 2017)." We also added extensive text and a new figure to the manuscript that resolve the seeming discrepancy between immission and emission maps. Additional to the original manuscript the validity of the present approach of connecting immission and emission of particulate pollution was tested by calculating temporal changes of eBC for subsets of back trajectories passing over two separate prominent emission regions, region A to the Northwest and B to the Southeast of the measuring stations. Consistent with reported emission data the calculated immission decreases over region A are significantly stronger than over region B.

**Comment:**

The authors state that NOx and SO2 maps look very similar (related to figures 1 and 2). I wonder why these results are shown in the paper, especially when considering that both these trace gases are important parts of trend analyses presented later in the paper.

Response: In a new supplement to the manuscript we display seasonal maps of ozone  $SO_2$ , and  $NO_3$ .

**Comment:**

Based on Figure 3, the authors discuss the influence of wind speed and wet scavenging on measured concentrations. Unfortunately, this discussion remains very qualitative. If included in this paper, more concrete results on the influence of these two variables should be presented.

Response: We agree with the reviewer concerning the qualitative nature of the criticized discussion. It is due to the fact that the HYSPLIT trajectory model only provides very limited meteorological information along the trajectories taken from neighboring grid-points of the used meteorological maps. Nevertheless, we feel that the maps provided in Fig. 3 provide a useful illustration of possible reasons for the discrepancies between the emission and the immission maps.

**Comment:**

In the last part of the paper, the authors investigate the time evolution of emissions during the past 10 years. I have a couple of concerns related to this. First, the authors only stated that they optimized the annual and monthly emissions of 2009 for the rest of the time series using their measurements. This is too scarce information. The authors should provide more details on how this exercise what done in practice. Response: See response to same comment by reviewer 1

**Comment:**

Second, as well known, emission changes have been very different in different parts of Europe during the past years. Is the approach applied in this paper able to catch this feature in any way, and if it is not, this should be explicitly mentioned in the paper.

Response: In the subsection of lines 294 through 322 of the original manuscript we take up this issue, extending our discussion of possible influences from neighboring countries all way to Romania.

**Comment:**

The final conclusion of this paper remain somewhat vague. The authors should more concrete summarize what new information this study brings on top of what is already known about the past emission changes in Europe. The authors mention some of the limitations of the current study yet, based on the comments give above, I feel that this list could be expanded a bit. Response: More specific information was added to the conclusions and the list of limitations of the study was expanded.

**Literature**

[revised manuscript text omitted]

year period 77516 hours with at least concurrent $PM_{10}$ -data at
all four stations could be utilized. |
|-----------------------------------------------------------------------------------------------------------------------------------------------------------------------------|

|------------------|-----------------------------------------------------------------------|
Tiefgestellt |
Tiefgestellt |
|                  | $\begin{tabular}{lllllllllllllllllllllllllllllllllll$                 |

[revised manuscript text omitted]

375 In Fig. 3 annual average emissions of PM10, BC, SO2, and NOx are mapped for 2009 according 376 to the EDGAR emission database. Except for the absolute numbers the maps for SO2, and NOx 377 Jook rather similar to those for particulate emissions. They all emphasize highly populated and 378 industrialized emissions center. Beyond that the SO2-map accentuates individual large 379 combustion sources such as conventional power plants. Whereas the strong emissions in 380 Northern Italy are seen in the maps of PM10, BC, and NOx emissions in the countries in the 381 Southeastern sector of the maps by no means reflect the high concentrations of particulate 382 components seen in the immission maps of Figs. 1 and 2.

383

384 The seeming discrepancy between the immission maps in Figs. 1 and 2 and the emission maps of Fig. 3 can be resolved. For that purpose, the EDGAR-emissions of PM10, BC, SO2, 385 386 and NOx along all 350593 hourly back trajectories to the four stations during the ten studied 387 years were summed up. Then the sums were extrapolated back along each trajectory. In Fig. 388 S4 10-year average maps of these extrapolated emission sums are displayed. As in Fig. 3 except 389 for the absolute numbers there is a strong similarity between the four mapped component sums. 390 Because of the integral nature of the mapped results one cannot expect the maps in Fig. S4 to 391 locate correctly specific emission centers. However, they certainly indicate the map sectors 392 from which the most substantial emissions could have reached the stations. As in Figs. 1 and 14

Current explanations of the new particle formation process (as indicated by N10-26) point towards photochemical processes that take place in plumes that contain sulfur dioxide (Größ et al., 2018). Several authors have stressed the possibility of particles to be formed in lofted layers, which are subsequently mixed to the ground (Platis, 2016), and/or in sulfur-rich plumes downstream of industrial point sources such as power plants (Junkermann and Hacker, 2018). ¶

-The trajectory extrapolated PM10-concentrations in Fig. 1 most strongly show the contrast between the relatively clean Northwest sector and the high concentrations in the Southeast sector of the maps.

[revised manuscript text omitted]

| [1] verschoben (Einfügung)                                                                                                                                           |
|----------------------------------------------------------------------------------------------------------------------------------------------------------------------|

| 548        | medians $\tilde{E}_{i=1,4}$ were formed according to Eq. 1 (exemplified for BC). Medians were chosen to                                                                                                      |                       |                                                                                                                                                                                           |
|------------|--------------------------------------------------------------------------------------------------------------------------------------------------------------------------------------------------------------|-----------------------|-------------------------------------------------------------------------------------------------------------------------------------------------------------------------------------------|
| 549        | reduce the effect of outliers due to local emission and scavenging events.                                                                                                                                   |                       |                                                                                                                                                                                           |
| 550        |                                                                                                                                                                                                              |                       |                                                                                                                                                                                           |
| 551        | $\tilde{E}_{BC} = Median(\sum_{n=1}^{121} E_{BC})_{m=1,M}$ Eq. 1                                                                                                                                             |                       |                                                                                                                                                                                           |
| 552        |                                                                                                                                                                                                              |                       |                                                                                                                                                                                           |
| 553        | The monthly median emission sums $\tilde{E}_{i=1,4}$ were modified with a monthly $(f_m)$ and an annual                                                                                                      | <                     | [1] nach oben verschoben: We expected both, seasonal variations and a long-term trend in the emissions.                                                                                   |
| 554        | factor $(g_y)$ in order to simulate respective median monthly measured concentrations taken over                                                                                                             |                       | Gelöscht: In order to optimize the                                                                                                                                                        |
| 555        | all stations. Thus, for each component 12 monthly and 10 annual trend factors determined the                                                                                                                 |                       |                                                                                                                                                                                           |
| 556        | agreement of modified summed emissions and measured concentrations. As objective or utility                                                                                                                  |                       |                                                                                                                                                                                           |
| 557        | function $\chi^2$ the sum of squared deviations between annually and monthly modified emission                                                                                                               |                       |                                                                                                                                                                                           |
| 558        | sums and monthly median measured concentrations was formed taken over the 120 months of                                                                                                                      |                       |                                                                                                                                                                                           |
| 559        | the present study (exemplified for BC in Eq. 2).                                                                                                                                                             |                       |                                                                                                                                                                                           |
| 560        |                                                                                                                                                                                                              |                       |                                                                                                                                                                                           |
| 561        | $\chi^{2}_{BC} = \sum_{j=1}^{120} (f_{m=1,12} \cdot g_{y=1,10} \cdot \tilde{E}_{BC} - eBC)^{2} $ Eq. 2                                                                                                       |                       | Formatiert: Tabstopps: 15 cm, Links                                                                                                                                                       |
| 562        |                                                                                                                                                                                                              |                       |                                                                                                                                                                                           |
| 564        | that optimized the12 monthly and 10 annual factors for each of the four measured components.                                                                                                                 |                       | Formatiert: Rechtschreibung und Grammatik prüfen                                                                                                                                          |
| 565        | We used Excel's ® implementation of the GRG-solver procedure for the optimization. After                                                                                                          |                       | Gelöscht: The GRG-solver minimizes the average absolute                                                                                                                                   |
| 566        | optimizing month and trend factors the average relative deviation between emission-simulated                                                                                                                 |                       | annual and 12 monthly adjustment factors at the summed
emissions                                                                                                                       |
| 567
568 | and measured monthly median curves is 14%, 21%, 25%, and 18% for $PM_{10}$ , eBC, SO 2 , and NO 2 and respectively. The optimized monthly median emission sums for all four parameters |                       | Gelöscht: was repeated for a fit of the trajectory-summed
emissions of PM 10 , BC, SO 2 , and NO x with the respective
measured time series |

[revised manuscript text omitted]

**given for each component**

|                   |                                                              |                                   |                 |                                         |                                   | PM10                                       | PM10                      |                                |                                       | $\underline{O_3^8}$ |
|-------------------|--------------------------------------------------------------|-----------------------------------|-----------------|-----------------------------------------|-----------------------------------|--------------------------------------------|---------------------------|--------------------------------|---------------------------------------|---------------------|
| Station           | Acronym                                                      | Latitude                          | Longitude       | MPSS 1                       | eBC 2                  |                                            |                           | NO x 6   | $SO_2^7$                              |                     |
|                   |                                                              |                                   |                 |                                         |                                   | continous 3,4                   | discontinous 5 |                                |                                       |                     |
| Collmberg         | СО                                                           | 51.3                              | 13              |                                         |                                   | 85054                                      |                           | 88838                   |                                       | 88792        |
| Melpitz           | ME                                                           | 51.5                              | 12.9            | 81561                            | 88196                      |                                            | 88822                     | 86260                          | 85541                          | 84421        |
| Neuglobsow        | NG                                                           | 53.1                              | 13              | 57962                            | 77540                             | 71202                                      |                           | 83718                          | 87778                          | 87943        |
| Waldhof           | WA                                                           | 52.8                              | 10.8            | 84276                            | 80725                      | 88321                               |                           | 85503                          | 82386                                 | 87373        |
| ▲ 1 MF | <mark>2SS_</mark> -scanning n
entific; 3 TEOM- | nobility particle
FDM - Tapere | e size spectrom | eter TROPOS (10 -
illating microbala | ⊥
-800 nm);²№
Ince fitted w | I
1AAP - Multi-ang
ith a filter dyna | l le absorption phot      | ometer 5012 T
ystem 1405 Tł | hermo Fischer
nermo Fischer |                     |
| Scie              | entific; ⁴SCHAR                                              | P - Synchronize                   | d Hybrid Ambie  | ent Real-time Part                      | iculate Moni                      | tor 5030 Thermo                            | Fischer Scientific;       | 5 HVS – High Vc     | lume Sampler                          |                     |
| DIG               | ITEL DH-80; 6 T                                   | LA-NOx –Trace                     | e Level NOx An  | alyzer 42i-TL The                       | rmo Fischer :                     | Scientific; 7 TLA-S             | 602 - Trace Level S       | O2 Analyzer 43                 | 3i-TLE Thermo                         |                     |
| Fisc              | her Scientific <mark>;</mark>                                | 3                                 |                 |                                         |                                   |                                            |                           |                                |                                       |                     |
|                   |                                                              |                                   |                 |                                         |                                   |                                            |                           |                                |                                       |                     |

1 157 Table 2 Median concentrations of eBC concentrations (µgm-3) and temporal trends (2009-2018) of eBC in terms of Sen-Theil slope (% per year) as

1158 determined for air masses passing over Regions A and B as analyzed at the stations Melpitz (ME), Neuglobsow (NG), and Waldhof (WA). For

1159 comparison the national annual decreases in BC emissions 2009-2017 in % according to the European Environmental Agency are added.

|                                | DELT        | [A]         |              |              |              | Med         | lian eB     | C in        |                         |                         |                  |                     | Decrease i        | in national BC | emissions       | Formatiert: Schriftart: (Standard) Times New                              |
|--------------------------------|-------------|--------------------|--------------|--------------|--------------|-------------|-------------|-------------|-------------------------|-------------------------|------------------|---------------------|-------------------|----------------|-----------------|---------------------------------------------------------------------------|
|                                | T*   | No .        | of ba        | ick traje    | ectories     |             | $\mu m/m^3$ |             | Sen                     | -Theil                  | slope i          | n % per year | in % per y | ear     |                 |                                                                           |
| A                              | in h | M                  | E            | NG           | WA    | ME   | NG   | WA   | ME               | NG               | WA        | 3 Stations** | Belgium    | Netherlands    | Germany         |                                                                           |
|                                |             | 1 219              | 941 1 | 17514 | 27218        | 0.38        | 0.40        | 0.41 | 6.40             | 6.80             | 4.80             | -5.85        | -6.1%      | 6.1%    | -4.9%    |                                                                           |
| D                              |             | 2 10               | 07 1         | 1 42 (0      | 00100        | 0.20        | 0.40        | 0.41        | =                       | =                       | -         | 5.00                |                   |                |                 | Formatiert: Schriftart: (Standard) Times New                              |
| Region A                       |             | 3 180              | 505 I | 14268        | 22132        | 0.38        | 0.40        | 0.41 | 6.40             | 6.90             | 4.80             | -5.89        |                   |                |                 | Formatiert: Schriftart: (Standard) Times New                              |
| B-NL-NRW                       |             | 6 14        | 302 1 | 10086        | 15936 | 0.39 | 0.40 | 0.42 | 6.40             | -
7.60        | 5.10      | -6.19        |                   |                |                 | Formatiert: Schriftart: (Standard) Times New                              |
|                                |             | 12 68              | 817          | 3746         | 6131         | 0.40        | 0.50        | 0.50        | -
7.10               | -
7.90        | -
5.30 | -6.62               |                   |                |                 | Formatiert: Schriftart: (Standard) Times New                              |
|                                |             |                    |              |              |              |             |             |             |                         |                         |                  |                     | Czech             |                |                 | Formatiert: Schriftart: (Standard) Times New                              |
|                                |             |                    |              |              |              |             |             |             |                         |                         |                  |                     | Rep.              | Poland  | Slovakia |                                                                           |
|                                |             |                    |              |              |              |             |             |             | Ξ                       | , E                     | , E              |                     |                   |                |                 |                                                                           |
|                                |             | 1 110              | )96   | 5264  | 4191  | 1.10 | 1.19 | 1.13 | 3.60             | 3.40                    | 1.70      | -3.16        | -2.8%      | 6 0.5%  | -2.3%    |                                                                           |
| Desire D                       |             | 2 0                | CO 1         | 4220         | 2541         | 1.00        | 1 10        | 1.10        | 2 40                    | 2 40                    | 2 10             | 2.14                |                   |                |                 | Formatiert: Schriftart: (Standard) Times New J                            |
| Region B                       |             |             | 501   | 4339         | 3541  | 1.08        | 1.18        | 1.12        | 3.40             | 3.40             | 2.10             | -3.14        |                   |                |                 | Formatiert: Schriftart: (Standard) Times New 1                            |
| CZ-PL-SK                       |             | 6 7         | )00   | 3062         | 2570  | 1.05 | 1.09 | 1.11        | -
4.00 | -
2.90        | 2.70      | -3.47               |                   |                |                 | Formatiert: Schriftart: (Standard) Times New                              |
|                                |             | 12 3 | 528   | 1410  | 1277  | 1.00 | 1.00 | 1.00 | -
3.70 | -
3.00 | -
2.70 | -3.34        |                   |                |                 | Formatiert: Schriftart: (Standard) Times New                              |
| ALL                            |             | 85                 | 846 7        | 75190        | 78356        | 0.45        | 0.36        | 0.36        | 5.90                    | 5.60                    | 4.00             | -5.18               |                   |                |                 | Formatiert: Schriftart: (Standard) Times New 1                            |
| Sun (2020)
Minimum t | ime en      | ent over           | the c        | mecifie      | ed source    | e regio     | n **\A      | aighte      | 4.40                    | 7.80                    | 3.20      | o the available p   | umber of back     | trajectories   |                 | Formatiert: Schriftart: (Standard) Calibri, 11 F
Schriftfarbe: Schwarz |
| IVIIIIIIIIIIIIIIIIIIII         |      |                    | the s        | peente       | u soure      | e regit     | , vv        | Cignite     | umear                   | 1, 000                  | i unig t         |                     |                   | trajectories.  |                 | Formationt: Finzug: Links: 0 cm Erste Zeile:                              |

Zeilenabstand: einfach

| 1161 | Table 3 Percenta                                                          | al decreases in th | ne anthro | opogeni   | c emissions of | ° PM 10 , BC, SC | D 2 , and |        | Gelöscht:Seite

---

## Referee Report (RR1)

Thanks for the revision of the manuscript. I think the manuscript has clearly improved. I have still one main concern that has not been resolved and that should be addressed prior to publication in ACP.

The methodical part of the back trajectory analysis in section 3 is still insufficient. There is very little or almost no information given how exactly the back trajectories are used for calculation of the immission maps. It seems that the emissions were summed up all along the trajectory without taking the height of the trajectory into account or whether the air parcel had contact with the PBL and a chance to pick up emissions or not. Although I am not an expert on back trajectory analysis, I would consider this issue very essential for the paper. For linking emissions with concentration maps (immission maps) only emissions from grid cells where the back trajectory was close to the ground should be considered. If this is not the case here, I would have doubts regarding the validity and the usefulness of the calculated immission maps and consequently of the key messages of this paper. So it is in my view crucial that the applied back trajectory analysis has been done correctly. This must be demonstrated by a more detailed and convincing description of the applied method.

---

## Author Response (AR2)

Reviewer comment:

*The methodical part of the back trajectory analysis in section 3 is still insufficient. There is very little or almost no information given how exactly the back trajectories are used for calculation of the immission maps. It seems that the emissions were summed up all along the trajectory without taking the height of the trajectory into account or whether the air parcel had contact with the PBL and a chance to pick up emissions or not. Although I am not an expert on back trajectory analysis, I would consider this issue very essential for the paper. For linking emissions with concentration maps (immission maps) only emissions from grid cells where the back trajectory was close to the ground should be considered. If this is not the case here, I would have doubts regarding the validity and the usefulness of the calculated immission maps and consequently of the key messages of this paper. So it is in my view crucial that the applied back trajectory analysis has been done correctly. This must be demonstrated by a more detailed and convincing description of the applied method.*

Response:

We fully agree with the reviewer as to the missing crucial information about our use of the trajectories.  We are very sorry that this information somehow got lost in the course of our revisions.  It was maintained though in the caption to the maps.

We now complemented section 3 with the crucial sentence:

[revised manuscript text omitted]

¶
¬Current explanations of the new particle formation process (as indicated by $N_{10-26}$) point towards photochemical processes that take place in plumes that contain sulfur dioxide (Größ et al., 2018). Several authors have stressed the possibility of particles to be formed in lofted layers, which are subsequently mixed to the ground (Platis, 2016), and/or in sulfur-rich plumes downstream of industrial point sources such as power plants (Junkermann and Hacker, 2018). ¶
¶
¬The trajectory extrapolated $PM_{10}$-concentrations in Fig. 1 most strongly show the contrast between the relatively clean Northwest sector and the high concentrations in the Southeast sector of the maps.

[revised manuscript text omitted]

Rechts:  2 cm, Unten:  2.5 cm, Breite:  29.66 cm, Höhe:  20.99 cm

| Seite 40: [3] Formatierte Tabelle | Nemo | 30.04.20 11:06:00 |
|---|---|---|

| Seite 40: [4] Gelöscht | Nemo | 01.04.20 11:49:00 |
|---|---|---|

| Seite 40: [4] Gelöscht | Nemo | 01.04.20 11:49:00 |
|---|---|---|

| Seite 40: [5] Formatiert | Nemo | 02.04.20 15:32:00 |
|---|---|---|

Schriftart: 12 Pt.

| Seite 40: [6] Formatiert | Nemo | 02.04.20 15:32:00 |
|---|---|---|

Schriftart: 12 Pt.

| Seite 40: [7] Formatiert | Nemo | 30.04.20 11:06:00 |
|---|---|---|

Tiefgestellt

| Seite 40: [7] Formatiert | Nemo | 30.04.20 11:06:00 |
|---|---|---|

Tiefgestellt

| Seite 40: [8] Formatiert | Nemo | 02.04.20 15:32:00 |
|---|---|---|

Schriftart: 12 Pt.

| Seite 40: [8] Formatiert | Nemo | 02.04.20 15:32:00 |
|---|---|---|

Schriftart: 12 Pt.

| Seite 40: [9] Formatiert | Nemo | 02.04.20 15:32:00 |
|---|---|---|

Schriftart: 10 Pt.

| Seite 40: [10] Formatiert | Nemo | 02.04.20 15:32:00 |
|---|---|---|

Schriftart: 10 Pt.

| Seite 40: [11] Formatiert | Nemo | 30.04.20 11:07:00 |
|---|---|---|

Hochgestellt

| Seite 40: [11] Formatiert | Nemo | 30.04.20 11:07:00 |
|---|---|---|

Hochgestellt